# The Forecast After the Forecast: A Post-Processing Shift in Time Series

**Daojun Liang** [1,2]   **Qi Li** [1,2, *]   **Yinglong Wang** [1,2, *]   **Jing Chen** [1,2,]   **Hu Zhang** [1,2]

**Xiaoxiao Cui** [3]   **Qizheng Wang** [1,2]   **Shuo Li** [4, 5]

[1] Key Laboratory of Computing Power Network and Information Security, Ministry of Education,
   Shandong Computer Science Center (National Supercomputing Center in Jinan),
   Qilu University of Technology (Shandong Academy of Sciences), Jinan, 250103, China
[2] Shandong Provincial Key Laboratory of Computing Power Internet and Service Computing,
   Shandong Fundamental Research Center for Computer Science, Jinan, 250103, China
[3] Joint SDU-NTU Centre for Artificial Intelligence Research (C-FAIR), Shandong University
[4] Department of Computer and Data Sciences, Case Western Reserve University, Cleveland, USA
[5] Department of Biomedical Engineering, Case Western Reserve University, Cleveland, USA
   `liangdj@sdas.org`, `{li.qi, wangyinglong}@qlu.edu.cn`
   `{chenj,zhanghu}@sdas.org`, `cuixiaoxiao711@hotmail.com`
   `574370699@qq.com`, `shuo.li11@case.edu`

## Abstract

Time series forecasting has long been dominated by advances in model architecture, with recent progress driven by deep learning and hybrid statistical techniques. However, as forecasting models approach diminishing returns in accuracy, a critical yet underexplored opportunity emerges: the strategic use of post-processing. In this paper, we address the last-mile gap in time-series forecasting, which is to improve accuracy and uncertainty without retraining or modifying a deployed backbone. We propose $\delta$-Adapter, a lightweight, architecture-agnostic way to boost deployed time series forecasters without retraining. $\delta$-Adapter learns tiny, bounded modules at two interfaces: input nudging (soft edits to covariates) and output residual correction. We provide local descent guarantees, $O(\delta)$ drift bounds, and compositional stability for combined adapters. Meanwhile, it can act as a feature selector by learning a sparse, horizon-aware mask over inputs to select important features, thereby improving interpretability. In addition, it can also be used as a distribution calibrator to measure uncertainty. Thus, we introduce a Quantile Calibrator and a Conformal Corrector that together deliver calibrated, personalized intervals with finite-sample coverage. Our experiments across diverse backbones and datasets show that $\delta$-Adapter improves accuracy and calibration with negligible compute and no interface changes.

## 1 Introduction

Time Series Forecasting (TSF) powers decisions across energy Anderson (1976), finance Hyndman & Athanasopoulos (2018), retail Piccolo (1990), and the sciences Gardner Jr (1985). Despite impressive gains from modern neural forecasters Ekambaram et al. (2024); Hollmann et al. (2025); Liang (2025); Liu et al. (2025), ranging from temporal convolutions Lea et al. (2016); Wu et al. (2019; 2022); Li et al. (2023) and Transformers Zhou et al. (2021); Nie et al. (2023); Wang et al. (2024a); Liu et al. (2023); Ye et al. (2024); Wang et al. (2024a); Liang et al. (2024b) to hybrid statistical–neural models Liu et al. (2025); Ekambaram et al. (2024), condition drift Baier et al. (2020); Liang et al. (2025) is still not alleviated. Conventional remedies, e.g., full fine-tuning, architectural changes, or ensembling, either demand substantial compute, risk destabilizing a hardened system, or complicate operations. To cope with this, testing-time adaptation (TTA) is introduced into TSF. The testing-time methods aim to mitigate test-time concept drift via linear adapter updates

---

*Corresponding author.

Kim et al. (2025), dynamic gating Grover & Etemad (2025), parallel forecaster combines Lee et al. (2025), layer-wise adjustment Pham et al. (2023), and dynamic model selection Wen et al. (2023). However, these methods rely, to varying degrees, on future labels for online model updates, thereby introducing label leakage, where future ground-truth labels are unavailable when actually applied, that causes model performance degradation Liang et al. (2024a); yee Ava Lau et al. (2025). Furthermore, LoRA-style adapters Hu et al. (2022); Pfeiffer et al. (2020); Li & Liang (2021) in NLP tend to lead to high performance variance, since the output range is not fixed Biderman et al. (2024).

Thus, TSF in real deployments still faces the last-mile gap: 1) Conditions drift Baier et al. (2020), which refers to gradual changes in the data-generating process (e.g., seasonal regime shifts, covariate shifts in demand patterns) that occur after the model has been deployed, making full retraining costly; 2) High performance variance. Existing post-processing techniques are prone to have high performance variance due to unstable training; 3) Inefficient training/inference. Using complex modules or frequent updates to absorb low-complexity residuals Vovk et al. (2017; 2018) makes models suffer from inefficient training/inference. Based on these, we ask a different question: *Can we really keep the strong forecaster intact and learn only a tiny, post-hoc module that makes small targeted corrections, so accuracy and reliability improve without heavy retraining?*

We answer "yes" with $\delta$-Adapter, a lightweight, model-agnostic framework that augments a frozen forecaster $F$ by learning a tiny adapter $A$ in two minimal placements: input-side nudging (softly editing covariates before inference) and output-side correction (residual refinement after inference). Concretely, we instantiate additive or multiplicative forms for both placements, with a small trust-region parameter $\delta \in (0, 1)$ that bounds edits for safety and stability. Since $A$ is a tiny network (e.g., shallow MLP or low-rank head) trained while $F$ remains frozen, it produces consistent gains with negligible training time and zero changes to $F$'s inference interface.

Further, a key instantiation of the input adapter is a feature-selector (mask) adapter that learns a sparse, nearly binary, horizon-aware mask $M \in [0, 1]^{L \times d}$ and applies it multiplicatively to the context $X' = X \odot M$. We train $M$ end-to-end with sparsity, temporal-smoothness, and budget regularizers so that the adapter preserves the base model's inductive biases while exposing the most consequential inputs for the frozen forecaster. This yields transparent selections, stable training, and strong empirical gains under tight compute budgets.

Beyond point accuracy, $\delta$-Adapter also upgrades forecast uncertainty without modifying $F$. We present two distributional correctors: 1) a Quantile Calibrator that learns horizon-wise quantile functions as bounded offsets from the point forecast, with a monotonic parameterization and pinball-loss training augmented by reliability regularization; and 2) a Conformal Calibrator that learns a scale function for normalized-residual conformal prediction, delivering finite-sample coverage with personalized, heteroscedastic intervals. Empirically, both calibrators achieve state-of-the-art coverage quality and produce tight, well-behaved intervals.

Through $\delta$-Adapter, this "last-mile" adjustment consistently improves forecasting accuracy in our experiments across diverse backbones and datasets, with negligible training time and no change to inference interfaces. The main contributions are:

- We formalize $\delta$-Adapter and instantiate two placements (input nudging and output residual correction) in additive/multiplicative forms, all drop-in and architecture-agnostic.
- We introduce a learnable, budgeted mask that identifies and preserves the most consequential inputs, improving transparency and stability.
- We propose quantile and conformal calibrators that deliver calibrated, heteroscedastic uncertainty with finite-sample coverage guarantees, all while keeping $F$ frozen.
- Across diverse backbones and benchmarks, $\delta$-Adapter improves accuracy and calibration; ablations illuminate the roles of $\delta$, capacity, horizon features, and residual structure.

## 2 METHODOLOGY

### 2.1 PROBLEM SETUP

Let $\mathcal{D} = \{(X^{(i)}, Y^{(i)})\}_{i=1}^N$ denote training pairs of context windows $X \in \mathbb{R}^{L \times d}$ and future targets $Y \in \mathbb{R}^{H \times m}$ (history length $L$, horizon $H$, $d$ covariates, $m$ target dimensions). A pre-trained

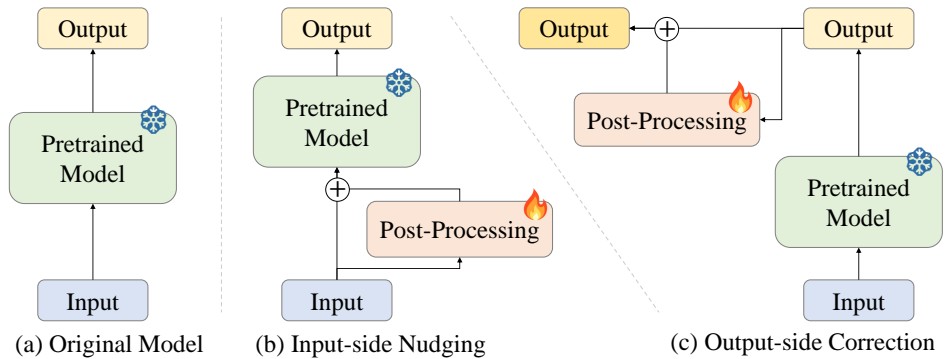

Figure 1: $\delta$-Adapter performs input nudging and output correction on the frozen forecaster.

forecaster $F$ maps $X$ to predictions $\hat{Y} = F(X) \in \mathbb{R}^{H \times m}$. We keep all parameters of $F$ fixed and introduce a lightweight, learnable adapter $A_\theta$ with parameters $\theta$ trained on $\mathcal{D}$. The adapter composes with $F$ via two families of edits:

$$\textbf{Input-side nudging:} \quad \tilde{X} = X + \delta\, A_\theta^{\text{in}}(X), \qquad \text{(additive input)} \qquad (1.1)$$

$$\tilde{X} = X \odot \left(1 + \delta\, A_\theta^{\text{in}}(X)\right), \qquad \text{(multiplicative input)} \quad (1.2)$$

$$\textbf{Output-side correction:} \quad \tilde{Y} = F(X) + \delta\, A_\theta^{\text{out}}(F(X), X), \qquad \text{(additive output)} \qquad (1.3)$$

$$\tilde{Y} = F(X) \odot \left(1 + \delta\, A_\theta^{\text{out}}(F(X), X)\right), \quad \text{(multiplicative output)} \ (1.4)$$

The base risk of $F$ under a loss $\ell$ is

$$\mathcal{R}(F) = \mathbb{E}_{(X,y) \sim \mathcal{D}} \left[ \ell\big(\tilde{Y}, Y\big) \right]. \tag{2}$$

Here, we consider two adapters, trained by minimizing empirical risk over $\theta$ with $F$ frozen, as shown in Eq. 1. The key questions are: (i) when does a small $\delta$ provably help; (ii) why do lightweight adapters suffice; and (iii) How do we choose $\delta$ and what is the stability of the adapter $A$? Now, let's answer these questions.

## 2.2 OUTPUT-SIDE ADAPTERS AS SHRINKAGE RESIDUAL LEARNING

Here, we consider the additive adapter, as shown in Eq. 1.3: $\tilde{Y} = F(X) + \delta A_\theta^{\text{out}}(F(X), X)$. With slight modifications, the relevant analyses and theories also apply to multiplicative adapters.

Let $r(X) = Y - F(X)$ denote the residual process. For squared error $\ell(\hat{Y}, Y) = \frac{1}{2}\|\hat{Y} - Y\|_2^2$, the population risk of the output adapter with a fixed $F$ equals

$$\mathcal{R}_{\text{out}}(\delta) = \tfrac{1}{2}\mathbb{E}\left[ \left\| r(X) - \delta g(X) \right\|_2^2 \right], \quad g(X) := A_\theta^{\text{out}}(F(X), X). \tag{3}$$

Expanding,

$$\mathcal{R}_{\text{out}}(\delta) = \tfrac{1}{2}\mathbb{E}\left[ \|r\|^2 \right] - \delta \underbrace{\mathbb{E}\left[\langle r, g \rangle\right]}_{\text{signal alignment}} + \tfrac{1}{2}\delta^2 \mathbb{E}\left[ \|g\|^2 \right]. \tag{4}$$

**Proposition 2.1** (Small-step improvement). *If* $\mathbb{E}[\langle r, g \rangle] > 0$*, then for all*

$$0 < \delta < \frac{2\mathbb{E}[\langle r, g \rangle]}{\mathbb{E}[\|g\|^2]}, \tag{5}$$

we have $\mathcal{R}_{\text{out}}(\delta) < \mathcal{R}_{\text{out}}(0) = \frac{1}{2}\mathbb{E}[\|r\|^2]$. The quadratic in $\delta$ has negative derivative at 0 and a unique minimizer $\delta^\star = \frac{\mathbb{E}[\langle r, g \rangle]}{\mathbb{E}[\|g\|^2]}$.

*Remark.* Improvement hinges on alignment between the learned correction $g$ and the residual $r$. Even when $A$ is tiny, if residuals have low-complexity structure (calendar offsets, horizon-dependent bias, scale drift), a small $g$ can achieve positive alignment, and a shrunken step $\delta$ guarantees risk reduction. This is exactly the first step of boosting with shrinkage or a stacked residual learner with a conservative learning rate.

In practice, we learn $g$ from finite data with a penalty $\Omega(\theta)$ (e.g., $\ell_2$, low rank, sparsity). The empirical objective

$$\min_\theta \tfrac{1}{2} \sum_i \left\| y_i - F(X_i) - \delta g_\theta(X_i) \right\|^2 + \lambda \Omega(\theta) \tag{6}$$

yields a shrunken projection of residuals onto the function class of $A$. With small $\delta$ and a low-capacity $A$, we target the dominant residual modes while avoiding variance blow-up.

## 2.3 Input-side adapters via first-order linearization

For the input-nudging adapter, as shown in Eq. 1.1: $\tilde{X} = X + \delta A_\theta^{\text{in}}(X)$, apply a first-order expansion of $F$ around $X$:

$$F\big(X + \delta u(X)\big) \approx F(X) + \delta J_F(X) u(X), \tag{7}$$

where $u(X) := A(X, h)$ and $J_F(X) \in \mathbb{R}^{H \times d}$ is the Jacobian of $F$ w.r.t. inputs. Under squared loss, replacing $g$ by $J_F u$ in the previous derivation yields

$$\mathcal{R}_{\text{in}}(\delta) \approx \tfrac{1}{2} \mathbb{E}\left[ \|r\|^2 \right] - \delta \mathbb{E}\left[ \langle r, J_F u \rangle \right] + \tfrac{1}{2} \delta^2 \mathbb{E}\left[ \|J_F u\|^2 \right]. \tag{8}$$

In general, for $\hat{y}_{\text{in}}(X; \delta) = F\big(X + \delta u(X)\big), \quad \mathcal{R}_{\text{in}}(\delta) = \tfrac{1}{2} \mathbb{E}\left[ \left\| y - \hat{y}_{\text{in}}(X; \delta) \right\|_2^2 \right]$, we have

**Proposition 2.2** (General $\delta$-step improvement). *If $\mathbb{E}[\langle r, J_F u \rangle] > 0$, then there exists $\delta > 0$ such that $\mathcal{R}_{\text{in}}(\delta) < \mathcal{R}_{\text{in}}(0)$ for all $\delta \in (0, \delta]$. And, if $F$ is affine in the near of $X$, Prop. 2.1 is also hold.*

The proof is given in Appendix B.2. This proposition states that for a differentiable loss $\ell$, the loss gradient w.r.t. inputs satisfies $\nabla_x \ell(F(X), y) = J_F(X)^\top \nabla_{\hat{y}} \ell$. Choosing $u(X) \approx -B \nabla_x \ell(F(X), y)$ for a small, learned preconditioner $B$ recovers a learned, damped gradient step in input space; training $A$ on data finds such steps implicitly without computing $J_F^\top$ at test time.

# 3 The stability of $\delta$-Adapter

## 3.1 Prediction stability under bounded input edits

Let $\tilde{X} = X + \delta A_\phi^{\text{in}}(X)$ (additive case). Then, we have

**Proposition 3.1** (Drift bound). *Assume the frozen forecaster $F$ is $L_F$-Lipschitz, the change in prediction is bounded by*

$$\|\tilde{Y} - \hat{Y}\| \le \delta L_F \|A_\phi^{in}(X)\| \le \delta L_F \sqrt{Ld}. \tag{9}$$

The proof is given in Appendix B.3. Further, let $\tilde{X} = X \odot \exp(\delta A_\phi^{\text{in}}(X))$, we have

**Corollary 1** (Multiplicative input edits). *If $\|X\|_\infty \le B_X$, then*

$$\|\tilde{Y} - \hat{Y}\| \le \delta e^\delta L_F B_X \|A_\phi^{in}(X)\|. \tag{10}$$

*In particular, for $\delta \le 1$, $\|\tilde{Y} - \hat{Y}\| = O(\delta)$.*

The proof is given in Appendix B.4. Corollary 1 means that small $\delta$ yields Lipschitz-stable prediction changes for input adapters.

## 3.2 Loss stability and guaranteed local improvement

Let $\hat{Y} = F(X)$ and consider an output edit $\tilde{Y} = \hat{Y} + \delta d$ with $d := A_\phi^{\text{out}}(\hat{Y}, X)$, we have

$$\ell(\tilde{Y}, y) \le \ell(\hat{Y}, y) + \delta \langle g, d \rangle + \frac{\beta}{2} \delta^2 \|d\|^2, \quad g := \nabla_u \ell(u, y)\big|_{u=\hat{Y}}. \tag{11}$$

If $d$ aligns with $-g$, i.e. $\langle g, d \rangle \le -\alpha \|g\| \|d\|$, we get

**Theorem 2** (Descent for output adapters). *If the per-sample prediction loss $\ell(\cdot, y)$ is $\beta$-smooth in its first argument (e.g., MSE, Huber), for any sample,*

$$\ell(\tilde{Y}, y) - \ell(\hat{Y}, y) \le -\delta \alpha \|g\| \|d\| + \frac{\beta}{2} \delta^2 \|d\|^2. \tag{12}$$

*Hence, for any $\delta \in \left(0, \frac{2\alpha\|g\|}{\beta\|d\|}\right)$, the loss strictly decreases. The optimal $\delta^\star = \frac{\alpha\|g\|}{\beta\|d\|}$ yields*

$$\ell(\tilde{Y}, y) - \ell(\hat{Y}, y) \leq -\frac{\alpha^2}{2\beta}\|g\|^2. \tag{13}$$

The proof is given in Appendix B.5.

*Remark.* With MSE, $g = \hat{Y} - y$, so the improvement is proportional to the squared residual magnitude. Further, with a bounded adapter family, the trained $A^{\text{out}}$ (minimizing batch loss) produces $d$ that correlates with $-g$ unless capacity is zero.

**Theorem 3** (Descent for input adapters). *Let $\tilde{X} = X + \delta v$ with $v := A^{in}_\phi(X)$. Assume $F$ is differentiable at $X$ with Jacobian $J_F(X)$. Define the effective prediction step $s := J_F(X)v$. Then for $\delta$ small,*

$$\ell(F(\tilde{X}), y) \leq \ell(\hat{Y}, y) + \delta\langle g, s\rangle + \frac{\beta}{2}\delta^2\|d\|^2 + O(\delta^2). \tag{14}$$

*If $\langle g, s\rangle \leq -\alpha\|g\|\|s\|$, there exists $\bar{\delta} > 0$ such that $\forall \delta \in (0, \bar{\delta})$ the loss strictly decreases. Moreover, optimizing the quadratic upper bound in $\delta$ yields the same margin as Theorem 2 up to $O(1)$ terms.*

The proof is given in Appendix B.6. Theorems 2 and 3 show that for sufficiently small $\delta$ and mild alignment, both adapter types reduce the loss locally, with explicit improvement margins.

## 3.3 COMPOSITIONAL STABILITY (INPUT + OUTPUT)

Let the full edit be $\tilde{X} = X + \delta v$, $\hat{Y}' = F(\tilde{X})$, then $\tilde{Y} = \hat{Y}' + \delta d(\hat{Y}', X)$. Under the same conditions as Prop. 3.1 and Theorems 2 and 3, we have:

**Proposition 3.2** (Composite drift and loss bound).

$$\|\tilde{Y} - \hat{Y}\| \leq \|\hat{Y}' - \hat{Y}\| + \delta\|d(\hat{Y}', X)\| \leq \delta L_F\|v\| + \delta C_d, \tag{15}$$

*so the model drift is $O(\delta)$. Further, for the loss,*

$$\ell(\tilde{Y}, y) \leq \ell(\hat{Y}, y) + \delta\langle g, s + d\rangle + \frac{\beta}{2}\delta^2\|s + d\|^2 + O(\delta^2), \tag{16}$$

The proof is given in Appendix B.7. If the combined step $s + d$ aligns with $-g$ by parameter-sharing or a learned gate, we inherit the same descent guarantee as Theorem 2.

## 4 IMPLEMENTATION

### 4.1 $\delta$-ADAPTER

$\delta$-Adapter targets structured residuals (bias, scale miscalibration, phase lag) while preserving $F$'s inductive biases. We encode this through three principles: 1) Boundedness: Enforce small edits via $\delta$ and penalties on $\|A_\theta(\cdot)\|$; 2) Low capacity: Use tiny architectures to avoid overfitting and respect production budgets. 3) Horizon awareness: Allow horizon-specific corrections without destabilizing temporal coherence. Concretely, we use a tiny MLP as the backbone and impose:

$$\|A^{\text{in}}_\theta(X)\|_\infty \leq 1, \quad \|A^{\text{out}}_\theta(\cdot)\|_\infty \leq 1, \tag{17}$$

via tanh squashing and optional clipping, so that $\delta$ is a direct bound on the maximum per-entry change. For multiplicative edits we ensure positivity where required by applying $\exp(\delta A_\theta(\cdot))$ as an alternative to $1 + \delta A_\theta$. For compositional adapters (input+output), as stated in Prop. 3.2, their parameters can be optimized in parallel during the training process.

### 4.2 FEATURE SELECTOR

A particularly transparent instantiation of our input adapter is to cast it as a learnable mask (selector) that selects the parts of the input that are most consequential for the frozen forecaster $F$. Concretely, for a context window $X \in \mathbb{R}^{L \times d}$, we parametrize an adapter $A_\theta$ that outputs a mask $M(X; \theta) \in [0, 1]^{L \times d}$, and apply it multiplicatively,

$$X' = X \odot M(X; \theta). \tag{18}$$

The mask is trained end-to-end while keeping $F$ fixed. Intuitively, $M$ plays the role of a soft selector: values near 1 keep information intact, values near 0 suppress it. To obtain discrete, human-readable selections without sacrificing differentiability, we employ relaxed Bernoulli parameterizations. Let $\alpha(X; \theta) \in \mathbb{R}^{L \times d}$ be adapter logits. We form a Gumbel-Sigmoid (Concrete) relaxation

$$M(X; \theta, \tau) = \sigma\Big(\frac{\log \alpha(X; \theta) + G}{\tau}\Big), \tag{19}$$

where $G$ is i.i.d. Gumbel noise, $\sigma(\cdot)$ is the logistic function, and $\tau > 0$ is a temperature annealed from a high value (smooth masks) to a low value (nearly binary). At inference, we may harden the mask via a threshold $M_{\text{hard}} = \mathbf{1}\{M > 0.5\}$ or keep it soft to avoid distributional brittleness. As a simpler alternative, we use a straight-through estimator: threshold in the forward pass, back-propagate through the corresponding sigmoid in the backward pass. Training the mask as a selector requires explicit structure in the objective. Given predictions $\tilde{Y} = F(X \odot M)$, we minimize

$$\min_\theta \underbrace{\mathcal{L}_{\text{pred}}(\tilde{Y}, Y)}_{\text{forecasting error}} + \lambda_1 \underbrace{\|M\|_1}_{\text{sparsity}} + \lambda_{\text{ent}} \underbrace{\sum H(M_{t,j})}_{\text{low entropy}} + \lambda_{\text{tv}} \underbrace{\text{TV}(M)}_{\text{temporal smoothness}} + \lambda_{\text{bud}} \underbrace{(\bar{m} - \kappa)_+}_{\text{budget}}, \tag{20}$$

where $\bar{m} = \frac{1}{Ld} \sum_{t,j} M_{t,j}$ is the average keep-rate and $\kappa \in (0, 1]$ is a user-specified budget, which stabilizes selection under correlations by constraining the feasible keep set, e.g., use at most 10% of inputs. The $\ell_1$ and entropy terms encourage sparse, nearly binary masks; the total-variation penalty $\text{TV}(M)$ promotes temporal contiguity, reflecting the fact that relevant patterns often span short intervals rather than isolated instants. See the specific expressions of each part in Appendix C.3.

## 4.3 DISTRIBUTION CALIBRATOR

Now, we introduce how to use the proposed adapter as a calibrator when the forecaster $F$ is frozen and produces only fixed-point predictions.

### 4.3.1 QUANTILE CALIBRATOR

If a distributional assumption is undesirable, the adapter can directly output horizon-wise quantiles as bounded offsets from the point forecast:

$$q_{\tau, \theta}(X) = \hat{Y} + \varepsilon a_\theta(X, \hat{Y}, \tau) \odot s_\theta(X, \hat{Y}), \tag{21}$$

where $a_\theta \in [-1, 1]^{H \times m}$ and $s_\theta > 0$ is a learned scale. To ensure monotonicity in $\tau$, we parameterize

$$q_{\tau_{j+1}, \theta} = q_{\tau_j, \theta} + \text{softplus}(d_{j,\theta}(X, \hat{Y})), \quad \tau_1 < \tau_2 < \cdots < \tau_J, \tag{22}$$

where $d_{j,\theta}$ is the adapter's raw increment for the gap between two adjacent quantile levels $\tau_j$ and $\tau_{j+1}$. Eq. 22 anchored at a central level (e.g., $\tau_{J/2}$) via the bounded offset around $\hat{Y}$. Then, for the training objective, we replace the point losses with pinball loss and add reliability regularization:

$$\min_\theta \frac{1}{N} \sum_{i=1}^N \sum_{j=1}^J \ell_{\tau_j}(Y^{(i)}, q_{\tau_j, \theta}(X^{(i)})) + \lambda_{\text{cal}} \mathcal{C}_{\text{rel}}(\theta) + \lambda_{\text{mag}} \|a_\theta\|_2^2. \tag{23}$$

where $\ell_\tau$ is the pinball loss; $\mathcal{C}_{\text{rel}}$ can be the same soft-coverage penalty as above, or a PIT-uniformity term computed by interpolating the predicted quantiles into a differentiable CDF and matching the PIT distribution to $\text{Uniform}(0, 1)$.

### 4.3.2 CONFORMAL CALIBRATOR

When strict distribution-free guarantees are needed, we combine a learned scale function with conformal prediction, i.e., we train $w_\theta(X, \hat{Y}) > 0$ (small adapter) to predict residual magnitude while keeping the mean at $\hat{Y}$:

$$\min_\theta \frac{1}{N} \sum_{i=1}^N \left|Y^{(i)} - \hat{Y}^{(i)}\right| / w_\theta(X^{(i)}, \hat{Y}^{(i)}) + \lambda \|w_\theta\|_2^2, \tag{24}$$

subject to a mild regularizer to keep $w_\theta$ near 1 on average. Then, we can use conformal scaling on a held-out calibration set $\mathcal{D}_{\text{cal}}$ to compute normalized residuals as

$$r^{(i)} = \frac{\left\|Y^{(i)} - \hat{Y}^{(i)}\right\|}{w_\theta\left(X^{(i)}, \hat{Y}^{(i)}\right)}, \quad (X^{(i)}, Y^{(i)}) \in \mathcal{D}_{\text{cal}}. \tag{25}$$

Then, the calibrated marginally valid prediction sets can be obtained by

$$\mathcal{C}_\alpha(X) = \left\{ y : \|y - \hat{Y}\| \leq \kappa_\alpha w_\theta(X, \hat{Y}) \right\}, \tag{26}$$

where $\kappa_\alpha$ is the empirical $(1-\alpha)$-quantile of $\{r^{(i)}\}$. This yields finite-sample coverage $1 - \alpha$ under exchangeability. The adapter $w_\theta$ personalizes interval width while $F$ remains untouched.

## 5 EXPERIMENTS

We validate the $\delta$-Adapter method on a variety of widely used datasets, see Appendix C.1. We test its gains when applied to pre-trained and state-of-the-art (SOTA) models (Section 5.1), its application as a feature selector (Section 5.2), and its effectiveness as an interval calibrator (Section 5.3). In this paper, we set $\delta = 0.1$ (0.01 for ETT datasets) and the learning rate of Adam to 1E-4, and conduct an ablation study on them at Section 5.4.

### 5.1 EFFECTIVENESS OF $\delta$-ADAPTER

Table 1: The improvement of $\delta$-Adapter on Pre-Trained models.

| Model | Sundial-S (Univariate) | | | | | | | | TTM-R2 (Multivariate) | | | | | | | |
|---|---|---|---|---|---|---|---|---|---|---|---|---|---|---|---|---|
| Type | original | | Ada-X | | | Ada-Y | | | original | | Ada-X | | | Ada-Y | | |
| Dataset | MSE | MAE | MSE | MAE | IMP | MSE | MAE | IMP | MSE | MAE | MSE | MAE | IMP | MSE | MAE | IMP |
| ELC | 0.427 | 0.463 | **0.334** | **0.410** | 17% | 0.404 | 0.451 | 4% | 0.180 | 0.272 | **0.167** | **0.262** | 6% | 0.168 | **0.262** | 5% |
| Traffic | 0.237 | 0.314 | **0.220** | **0.301** | 6% | 0.224 | 0.302 | 5% | 0.517 | 0.344 | **0.492** | 0.329 | 5% | **0.492** | 0.325 | 5% |
| Exchange | 0.249 | 0.332 | 0.241 | 0.332 | 2% | **0.235** | **0.329** | 3% | 0.094 | 0.213 | **0.090** | **0.206** | 3% | 0.092 | 0.210 | 1% |
| Weather | 0.427 | 0.463 | **0.025** | **0.005** | 96% | 0.039 | 0.059 | 89% | 0.150 | 0.196 | 0.148 | 0.193 | 2% | **0.143** | **0.191** | 4% |
| ETTm1 | 0.121 | 0.217 | **0.078** | **0.190** | 24% | 0.087 | 0.202 | 18% | 0.338 | 0.357 | 0.329 | 0.357 | 1% | 0.331 | 0.353 | 3% |
| ETTm2 | 0.348 | 0.420 | **0.201** | **0.325** | 32% | 0.254 | 0.371 | 19% | 0.177 | 0.259 | **0.174** | 0.243 | 4% | 0.175 | **0.240** | 4% |

We first verify the performance gains of $\delta$-Adapter on pre-trained models, including Sundial-S (Univariate) Liu et al. (2025) and TTM-R2 (Multivariate) Ekambaram et al. (2024). The experimental results in Table 1 show that $\delta$-Adapter consistently enhances forecasting performance across all datasets and backbone models, confirming its effectiveness and generality. Both the input adapter (Ada-X) and the output adapter (Ada-Y) have achieved significant performance gains. These results highlight that training lightweight adapters while keeping the backbone frozen is a powerful and efficient way to boost predictive accuracy.

Table 2: Comparison of various adapter methods and online methods (averaged across all lengths).

| Model | DistPred | | | | iTransformer | | | | Autoformer | | | | Others | |
|---|---|---|---|---|---|---|---|---|---|---|---|---|---|---|
| Dataset | Offline | SOLID | TAFAS | LoRA | Ada-X+Y | Offline | SOLID | TAFAS | LoRA | Ada-X+Y | Offline | SOLID | TAFAS | Ada-X+Y | OneNet[†] | FSNet[†] |
| ELC | 0.182 | 0.182 | 0.182 | 0.180 | **0.175** | 0.190 | 0.190 | 0.190 | 0.186 | **0.180** | 0.515 | 0.502 | 0.510 | **0.478** | 0.417 | 0.537 |
| ETTh1 | 0.461 | 0.460 | 0.476 | 0.454 | **0.451** | 0.454 | 0.458 | 0.477 | 0.448 | **0.449** | 0.593 | 0.589 | 0.591 | **0.577** | 0.618 | 0.877 |
| ETTh2 | 0.390 | 0.391 | 0.402 | 0.385 | **0.379** | 0.388 | 0.393 | 0.448 | 0.384 | **0.377** | 0.438 | 0.435 | 0.436 | **0.426** | 0.581 | 0.587 |
| ETTm1 | 0.412 | 0.406 | 0.411 | 0.407 | **0.396** | 0.417 | 0.414 | 0.420 | 0.414 | **0.403** | 0.664 | 0.661 | 0.638 | **0.597** | 0.548 | 0.851 |
| ETTm2 | 0.285 | 0.285 | 0.288 | 0.281 | **0.274** | 0.300 | 0.298 | 0.304 | 0.293 | **0.290** | 0.339 | 0.339 | 0.338 | **0.321** | 1.171 | 1.113 |
| Exchange | 0.350 | 0.347 | 0.363 | 0.346 | **0.297** | 0.383 | 0.376 | 0.392 | 0.376 | **0.316** | 0.509 | 0.491 | 0.495 | **0.465** | 0.647 | 0.878 |
| Traffic | 0.453 | 0.453 | 0.455 | 0.449 | **0.440** | 0.475 | 0.475 | 0.476 | 0.468 | **0.461** | 0.972 | 0.959 | 0.975 | **0.942** | 0.567 | 0.701 |
| Weather | 0.256 | 0.255 | 0.256 | 0.251 | **0.242** | 0.259 | 0.257 | 0.259 | 0.255 | **0.244** | 0.325 | 0.316 | 0.325 | **0.299** | 0.390 | 0.541 |

[†] OneNet and FSNet are implemented based on the public library provided in their paper with no label leakage. For more details, please refer to Table 10 in Appendix C.10.

Then, we compared the proposed $\delta$-Adapter with other adapter methods and online learning methods by removing label leakage Liang et al. (2024a); yee Ava Lau et al. (2025). It is worth noting that when removing label leakage, some methods have a certain degree of performance degradation. This may be because the design of these methods relies excessively on future true values. Table 2 shows that the $\delta$-adapter achieves the lowest error on every dataset across all three backbones. The gains are sizeable on challenging sets, while remaining consistent on the ETT variants. Moreover, when

Table 3: Gains of $\delta$-Adapter on SOTA models (averaged across all lengths. See Table 8 for details).

| Model | DistPred | | | iTransformer | | | FourierGNN | | | FreTS | | | Autoformer | | |
|---|---|---|---|---|---|---|---|---|---|---|---|---|---|---|---|
| Dataset | Original | Ada-X | Ada-Y | Original | Ada-X | Ada-Y | Original | Ada-X | Ada-Y | Original | Ada-X | Ada-Y | Original | Ada-X | Ada-Y |
| ELC | 0.182 | 0.178 | **0.169** | 0.190 | 0.187 | **0.181** | 0.267 | 0.255 | **0.241** | 0.209 | 0.203 | **0.194** | 0.515 | 0.488 | **0.450** |
| Exchange | 0.350 | **0.302** | 0.319 | 0.383 | **0.348** | 0.349 | 0.380 | 0.393 | **0.379** | 0.416 | **0.412** | 0.422 | 0.509 | 0.481 | **0.462** |
| Traffic | 0.453 | 0.448 | **0.442** | 0.475 | 0.470 | **0.461** | 0.777 | 0.749 | **0.740** | 0.596 | 0.590 | **0.572** | 0.972 | 0.959 | **0.918** |
| Weather | 0.256 | 0.251 | **0.245** | 0.259 | 0.249 | **0.245** | 0.255 | 0.251 | **0.244** | 0.255 | 0.249 | **0.243** | 0.325 | 0.306 | **0.299** |
| ETTh1 | 0.461 | **0.457** | 0.458 | 0.454 | **0.453** | 0.456 | 0.561 | 0.546 | **0.542** | 0.482 | 0.474 | **0.471** | 0.593 | 0.583 | **0.577** |
| ETTh2 | 0.390 | **0.386** | 0.387 | 0.388 | **0.385** | 0.390 | 0.545 | **0.499** | 0.506 | 0.537 | **0.492** | 0.498 | 0.438 | 0.420 | 0.423 |
| ETTm1 | 0.412 | **0.399** | 0.402 | 0.417 | 0.407 | **0.406** | 0.456 | **0.447** | 0.447 | 0.405 | **0.401** | 0.401 | 0.664 | **0.604** | 0.637 |
| ETTm2 | 0.285 | **0.279** | 0.282 | 0.300 | **0.292** | 0.293 | 0.445 | **0.386** | 0.439 | 0.335 | **0.285** | 0.323 | 0.339 | 0.316 | 0.320 |

contrasted with OneNet and FSNet, $\delta$-Adapter paired with standard backbones yields substantially lower errors on all datasets, underscoring its plug-and-play effectiveness and robustness.

Next, we verify whether the $\delta$-Adapter provides gains to the SOTA forecaster. Table 3 shows that $\delta$-Adapter provides consistent and significant improvements across multiple SOTA models. For nearly all datasets, Ada-X and Ada-Y lead to lower prediction errors compared to the original models, demonstrating that the proposed adapters generalize well to diverse forecasting architectures. Notably, Ada-X again delivers the largest gains, particularly on challenging datasets such as Exchange, Traffic, and ETT series, confirming that refining the input signals before model inference is the most impactful strategy. Also, $\delta$-Adapter yields clear benefits, highlighting its plug-and-play nature and ability to enhance high-performing models. These results further validate that $\delta$-Adapter is a broadly applicable, efficient, and effective enhancement method for modern forecaster.

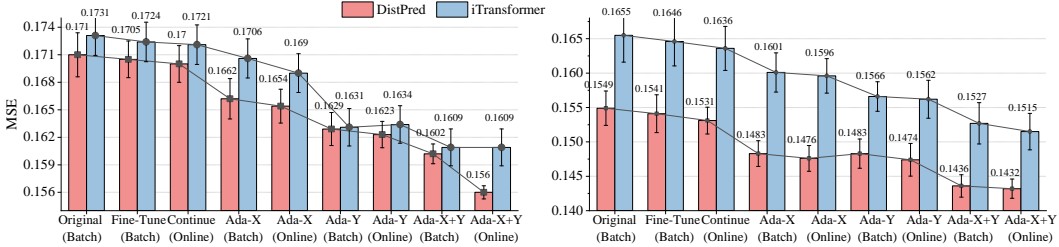

Figure 2: Performances of the forecaster $F$ and $\delta$-Adapter under batch or online training.

Finally, we test whether $\delta$-Adapter is effective under different compositions and training methods. Implementation and training details of Ada-X+Y are in Appendix C.5). Figure 2 shows that $\delta$-Adapter consistently reduces error under batch and online training. Each single adapter improves over the frozen forecaster and also outperforms conventional fine-tuning or continue-training. And training the adapters online yields further gains over batch. Importantly, Ada-X+Y delivers the lowest MSE in all settings, indicating robust and statistically reliable improvements.

## 5.2 EFFECTIVENESS OF THE FEATURE SELECTOR

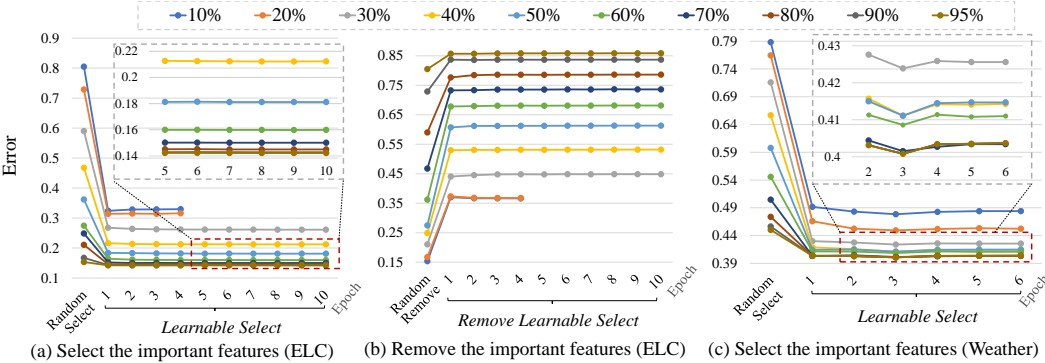

(a) Select the important features (ELC)  (b) Remove the important features (ELC)  (c) Select the important features (Weather)

Figure 3: Changes of forecaster's performance after selecting or removing valid features.

To verify the effectiveness of the mask adapter as a feature selector, we visualized its training process, as shown in Figure 3. It demonstrates that

Table 4: Best performance of the mask adapter and its mask ratio.

| Dataset | ELC | ETTh1 | ETTh2 | ETTm1 | ETTm2 | Traffic | Weather | Exchange |
|---|---|---|---|---|---|---|---|---|
| Original | 0.163 | 0.390 | 0.296 | 0.345 | 0.182 | 0.444 | 0.173 | 0.099 |
| Masked | 0.159 | 0.382 | 0,291 | 0.334 | 0.176 | 0.436 | 0.171 | 0.093 |
| Mask Ratio | 97% | 96% | 95% | 97% | 96% | 98% | 96% | 92% |

a learnable mask adapter reliably identifies the most informative input features under varying sparsity budgets. In subfigure (a), selected features yields markedly lower errors than random selection across all retention rates (10–95%) and converges within a few epochs. Conversely, when the learned features are removed (b), the forecaster's error rises substantially, often worse than removing an equal number of randomly chosen features. This shows that these features are uniquely critical to performance rather than incidental. Table 4 shows the mask ratio of the mask adapter when the best performance is achieved (no budget added), and Figure 4 visualizes important features in different proportions (most important features remain unchanged). These confirming that the learned selections consistently outperform random picks, and removing them degrades accuracy the most.

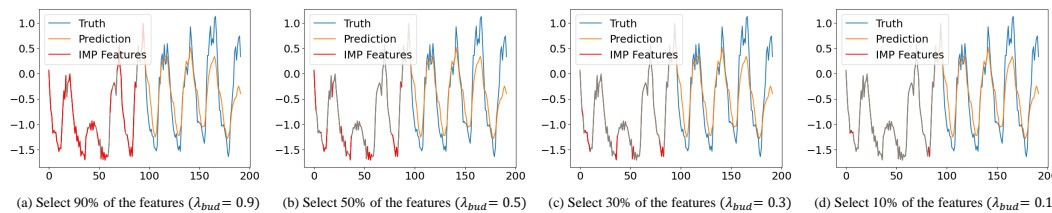

(a) Select 90% of the features ($\lambda_{bud}$= 0.9)   (b) Select 50% of the features ($\lambda_{bud}$= 0.5)   (c) Select 30% of the features ($\lambda_{bud}$= 0.3)   (d) Select 10% of the features ($\lambda_{bud}$= 0.1)

Figure 4: Visualization of different important features learned by the mask adapter.

## 5.3 PERFORMANCE OF CALIBRATOR

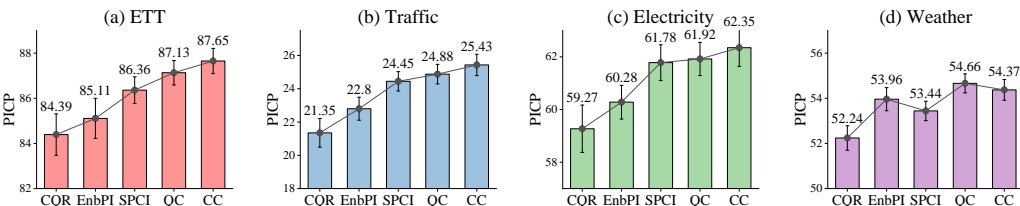

Figure 5: Comparisons among the Quantile (QC), Conformal (CC) calibrators and others.

Now, we verify the effect of $\delta$-Adapter as the Quantile Calibrator (QC) and Conformal Calibrator (CC). As shown in Figure 5, our calibrators consistently deliver the highest PICP, indicating better coverage reliability than strong baselines (CQR Romano et al. (2019), EnbPI Xu & Xie (2021), SPCI Xu & Xie (2023)). Further, in Figure 6, we illustrate that both calibrators produce well-calibrated intervals that expand near peaks and usually enclose the ground truth. QC tends to yield slightly wider, more conservative bands, while CC delivers comparably high coverage with tighter intervals.

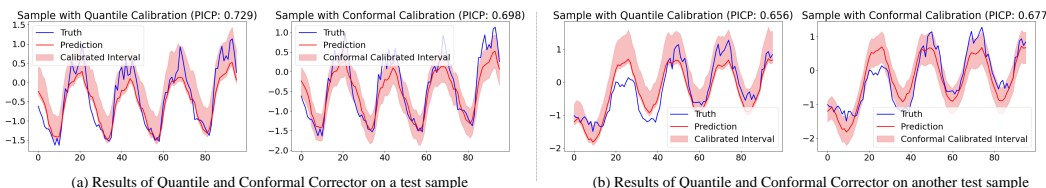

(a) Results of Quantile and Conformal Corrector on a test sample   (b) Results of Quantile and Conformal Corrector on another test sample

Figure 6: Visualization of the Quantile and Conformal calibrator predictions.

## 5.4 ABLATION STUDIES

The impact of $\delta$ is important. Figure 7 indicates that all adapter variants reduce error versus the frozen model, but the combined adapter (Ada-X+Y) delivers the lowest median errors and the tightest variability. Across placements, a moderate adjustment size is the most reliable, e.g., pushing to $\delta = 0.2$ yields smaller or inconsistent gains, suggesting overly aggressive corrections. It is confirmed that composing input and output adapters with a modest multiplicative trust-region produces the most accurate and stable forecasts.

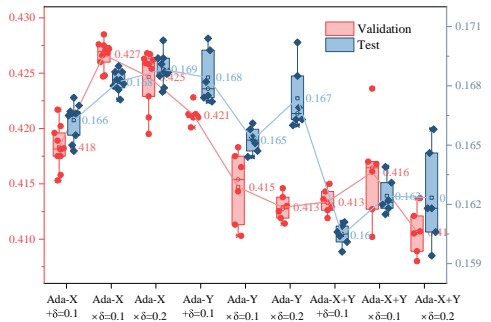

Figure 7: Performance of $\delta$-Adapter's variants.

Then, we used PatchTST Nie et al. (2023) and TimeMixer Wang et al. (2024a) as backbones to compare the performance of additive and multiplicative composite $\delta$-Adapter. As shown in Table 5, after adding the $\delta$-Adapter to PatchTST and TimeMixer, their performance has been significantly improved. the additive and multiplicative adapters reduce the MSE of PatchTST by 5.6% and 5.1% respectively across various datasets. However, for TimeMixer, the MSE reductions from the additive and multiplicative adapters are 1.6% and 1.8% respectively. This indicates that both have their respective advantages, and the increase of the additive adapter is relatively more significant. In addition, for efficiency of the $\delta$-Adapter, see Appendix C.8.

Table 5: Comparison of additive and multiplicative composite $\delta$-Adapter.

|  | PatchTST | | + Ada-X+Y | | + Ada-X×Y | | TimeMixer | | + Ada-X+Y | | + Ada-X×Y | |
|---|---|---|---|---|---|---|---|---|---|---|---|---|
|  | MSE | MAE | MSE | MAE | MSE | MAE | MSE | MAE | MSE | MAE | MSE | MAE |
| ELC | 0.167 | 0.252 | **0.159** | **0.245** | **0.159** | 0.246 | 0.145 | 0.243 | **0.143** | **0.241** | **0.143** | **0.241** |
| Weather | 0.178 | 0.219 | **0.161** | **0.220** | 0.165 | 0.224 | 0.168 | 0.216 | 0.166 | **0.214** | 0.164 | **0.214** |
| Traffic | 0.463 | 0.297 | 0.451 | 0.292 | **0.448** | **0.290** | 0.475 | 0.317 | **0.465** | **0.307** | 0.467 | 0.310 |

Finally, we investigated the impact of two key factors: $\delta$ and its learning rate. As shown in Figure 8, despite the large variation ranges of $\delta$ and the learning rate, the forecaster (iTransformer) can still maintain relatively stable prediction performance. However, other models, such as those that attempt to fine-tune pre-trained large models using Lora Hu et al. (2022), not only exhibit large variance but also lead to degradation in performance, which remains a problem worthy of further exploration.

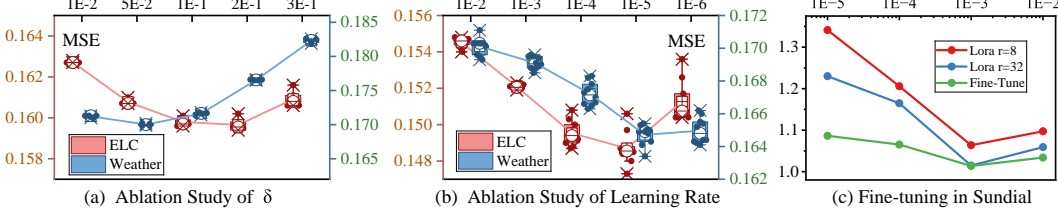

Figure 8: Variances of Ada-X with different $\delta$(a) and learning rates (b) and fine-tuned Sundial (c).

## 6 CONCLUSION

We present $\delta$-Adapter, a lightweight and post-hoc framework that improves frozen forecasters via bounded input nudges and output residual corrections. we provide theory guaranteeing local descent and stable composition. To enhance interpretability and robustness, we introduce a feature-selector adapter that learns a sparse, horizon-aware mask under budget priors, exposing the most consequential inputs while constraining edits. Beyond point forecasts, we deliver calibrated uncertainty via two distributional correctors: a Quantile Calibrator that learns quantile offsets trained with pinball loss, and a Conformal Corrector that estimates heteroscedastic scales for normalized-residual conformal prediction, yielding finite-sample coverage with personalized intervals. Across diverse backbones and datasets, $\delta$-Adapter yields consistent accuracy and calibration gains.

## ACKNOWLEDGEMENTS

This work was supported by Project of Key R&D Program of Shandong Province, China (2025CXPT095, 2025CXGC010107), Taishan Scholars Program: NO.tspd20240814, the Pilot Project for the Integration of Science, Education and Industry of Qilu University of Technology (Shandong Academy of Sciences) (2025ZDZX01), the "20 Articles" Project for Universities in Jinan (202534093), the Pilot Project for Integrated Innovation of Science, Education, the Industry of Qilu University of Technology (Shandong Academy of Sciences) (2024ZDZX08), the Taishan Scholars Program of Shandong Province (No. tsqnz20250747), and the National Natural Science Foundation of China (No. 62502250).

## ETHICS STATEMENT

Biases in benchmark creation: The authors are aware of the potential for bias in the creation of our benchmark entries. The selection and definition of dark patterns, as well as the design of benchmark prompts, may inadvertently refect the authors' perspectives and biases. This includes assumptions about user interactions and model behaviors that may not be universally accepted or relevant.

Misuse potential: While our intention with this benchmark is to identify and reduce the presence of dark design patterns in LLMs, we acknowledge the potential for misuse. There is a risk that malicious actors could use this benchmark to fine-tune models in ways that intentionally enhance these dark patterns, thereby exacerbating their negative impact.

## REPRODUCIBILITY STATEMENT

The code used in this paper can be found here. And we use notebooks to write some simple examples so that readers can quickly implement the results of the paper. The steps to reproduce the paper are:

- 1. Download the code.
- 2. Install the necessary environment.
- 3. Run "bash run.sh".
- 4. Or, run the provided notebook.

The code is given in this **Repository**.

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

# A    RELATED WORK

## A.1    CLASSICAL MODELS FOR TS FORECASTING

TS forecasting is a classic research field where numerous methods have been invented to utilize historical series to predict future missing values. Early classical methods Piccolo (1990); Gardner Jr (1985) are widely applied because of their well-defined theoretical guarantee and interpretability. For example, ARIMA Piccolo (1990) initially transforms a non-stationary TS into a stationary one via differencing, and subsequently approximates it using a linear model with several parameters. Exponential smoothing Gardner Jr (1985) predicts outcomes at future horizons by computing a weighted average across historical data. In addition, some regression-based methods, e.g., random forest regression (RFR) Liaw et al. (2002) and support vector regression (SVR) Castro-Neto et al. (2009), etc., are also applied to TS forecasting. These methods are straightforward and have fewer parameters to tune, making them a reliable workhorse for TS forecasting. However, their shortcoming is insufficient data fitting ability, especially for high-dimensional series, resulting in limited performance.

## A.2    DEEP MODELS FOR TS FORECASTING

The advancement of deep learning has greatly boosted the progress of TS forecasting. Specifically, convolutional neural networks (CNNs) LeCun et al. (1998) and recurrent neural networks (RNNs) Connor et al. (1994) have been adopted by many works to model nonlinear dependencies of TS, e.g., LSTNet Lai et al. (2018) improve CNNs by adding recursive skip connections to capture long- and short-term temporal patterns; DeepAR Salinas et al. (2020) predicts the probability distribution by combining autoregressive methods and RNNs. Several works have improved the series aggregation forms of Attention mechanism, such as operations of exponential intervals adopted in LogTrans Li et al. (2019), ProbSparse activations in Informer Zhou et al. (2021), frequency sampling in FED-former Zhou et al. (2022) and iterative refinement in Scaleformer Shabani et al. (2022). Besides, GNNs and Temporal convolutional networks (TCNs) Lea et al. (2016) have been utilized in some methods Wu et al. (2019); Li et al. (2023); Liu et al. (2022a); Wu et al. (2022) for TS forecasting on graph data. And some methods combine data-driven approaches with physical laws Hong et al. (2023; 2026) to achieve dynamic prediction of complex systems Hong et al. (2022). The aforementioned methods solely concentrate on the forms of aggregating input series, overlooking the challenges posed by the concept drift problem.

## A.3    TRANSFORMER-LIKE MODELS

Since TS exhibit a variety of patterns, it is meaningful and beneficial to decompose them into several components, each representing an underlying category of patterns that evolving over time Anderson (1976). Several methods, e.g., STL Cleveland et al. (1990), Prophet Taylor & Letham (2018) and N-BEATS Oreshkin et al. (2019), commonly utilize decomposition as a preprocessing phase on historical series. There are also some methods, e.g., Autoformer Wu et al. (2021), FEDformer Zhou et al. (2022), Non-stationary Transformers Liu et al. (2022b) and DistPred Liang (2025), that harness decomposition into the Attention module. The aforementioned methods attempt to apply decomposition to input series to enhance predictability, reduce computational complexity, or ameliorate the adverse effects of non-stationarity. Nevertheless, these prevalent methods are susceptible to significant concept drift when applied to non-stationary TS.

Furthermore, there are four themes that use deep learning to predict time series: (1) smarter transformers Vaswani et al. (2017), such as PatchTST Nie et al. (2022), iTransformer Liu et al. (2023), BasisFormer Ni et al. (2023), Minusformer Liang et al. (2024b), TimeXer Wang et al. (2024b), and Periodformer Liang et al. (2024c) which restructure attention or add learnable bases to extend context length, cut computation and boost accuracy; (2) competitive non-transformer backbones, including N-HiTS (hierarchical MLP) Challu et al. (2023), DLinear Zeng et al. (2023), PGN Jia et al. (2024), PSLD Liang et al. (2024d) and state-space models like TSMamba Ma et al. (2024), TimeMachine Ahamed & Cheng (2024) and FLDMamba Zhang et al. (2025), which deliver linear-time inference and rival or surpass transformers on long horizons Liang et al. (2023); (3) foundation-model initiatives, TimeGPT Garza et al. (2023), OneFitAll Zhou et al. (2023), TimeLLM Jin et al. (2024), UniTime Liu et al. (2024) and DAM Darlow et al. (2024) that pre-train on massive heterogeneous

corpora and achieve impressive zero-shot or few-shot performance across domains; and (4) training and interpretability advances, such as frequency-adaptive normalization (FAN) Ye et al. (2024), e.g., FreTS Yi et al. (2024b), FilterNet Yi et al. (2024a), and decomposition-aware architectures Yu et al. (2025b), e.g., which tackle non-stationarity, quantify uncertainty and make forecasts more transparent. ability to directly model the correlations among attributes within a series by decoupling the dependency on input length.

### A.4 ONLINE LEARNING

Online learning strategies embed concept drift adaptation Yu et al. (2025a) within the forecasting models themselves. One example is FSNet c, which leverages complementary learning systems theory to pair a slow-learning base forecaster with fast-adapting components. Another line of work is OneNet Wen et al. (2023), an online ensembling approach that dynamically combines two neural models: one specializes in capturing temporal dependencies within each series, and the other focuses on cross-series (covariate) relationships. Each of these deep learning techniques illustrates how integrating drift-awareness (through dual-model architectures, ensembling, or proactive adjustment) can improve TS forecasting performance in online Yu et al. (2024).

### A.5 POST-PROCESSING METHODS IN TIME SERIES

Testing-Time adaption (TTA) is very important for Time Series Forecasting. The adapter-based methods include SOLID Chen et al. (2024), TAFAS Kim et al. (2025) and its follow-ups PETSA Medeiros et al. (2025) and DynaTTA Grover & Etemad (2025), ELF Lee et al. (2025), etc., and online approaches, e.g., FSNet Pham et al. (2023) and OneNet Wen et al. (2023), aim to mitigate test-time concept drift. Specifically, SOLID retrains selected predictor layers using the most recent similar samples; TAFAS updates linear adapters by online detection of temporal cycles; PETSA and DynaTTA extends TAFAS with additional losses and dynamic gating to further enhance adaptability. These methods are either based on linear adapters, parallel fusion, or overall fine-tuning; and, they do not consider the impact of label leakage Liang et al. (2024a); yee Ava Lau et al. (2025). On the contrary, $\delta$-Adapter can perform non-linear adaptation on both input and output, with good theoretical guarantees. And it only relies on the most recent sample for fast updates. In addition, it can be used as a feature selector or a corrector.

### A.6 POST-PROCESSING METHODS IN NLP

Our work is conceptually related to the general parameter-efficient adaptation methods that have been developed primarily in NLP. Adapter modules for BERT and other Transformers add small task-specific bottleneck layers between pre-trained weights, keeping the backbone frozen while achieving near–fine-tuning performance on many downstream tasks Houlsby et al. (2019). This idea has been extended to multilingual and multi-task settings (e.g., MAD-X), where language and task adapters are stacked to enable cross-lingual transfer Pfeiffer et al. (2020). A complementary direction is low-rank adaptation (LoRA), which inserts trainable low-rank matrices into attention and feed-forward projections to adapt large language models with only a small number of additional parameters Hu et al. (2022). Another family of methods performs input-side adaptation via prompt and prefix tuning: instead of changing internal weights, they learn continuous prompts or prefixes at the embedding level that condition a frozen language model for each task Li & Liang (2021).

Compared with the above methods, $\delta$-Adapter adopts the same high-level principle of learning a small $\delta$-module around a frozen backbone, but it is tailored to TSF and operates strictly at the input/output interface of a possibly black-box forecaster. Specifically, we introduce horizon-aware input adapters, feature-masking modules, and output-side uncertainty adapters, and we analyze their behavior through Lipschitz-style stability and descent guarantees. To our knowledge, such an I/O level, theoretically characterized adapter framework for multi-horizon forecasting is not present in the existing NLP adapter or prompt-tuning literature, which primarily modifies internal layers or token embeddings of language models.

## A.7 CONFORMAL PREDICTION

Conformal prediction presents an alternative framework for distribution prediction, diverging from traditional parametric approaches. In their study, the authors in (Vovk et al., 2017; 2018) introduced a random prediction system and proposed a nonparametric prediction method grounded in conformal assumptions. By integrating conformal prediction with quantile regression in (Romano et al., 2019; Xu & Xie, 2021; 2023), they developed a method for constructing prediction intervals for the response variable. However, the practical application of conformal prediction is not without limitations. Its effectiveness is often constrained by the assumption of exchangeability of residuals, which may not hold in all contexts, particularly in the presence of temporal dependencies. This limitation can lead to less reliable prediction intervals when applied to non-independent and identically distributed (non-i.i.d.) data, thereby challenging its robustness in real-world scenarios where data often exhibit complex dependencies.

## A.8 TERMINOLOGY EXPLANATION

**Conditions drift**, which refers to gradual changes in the data-generating process (e.g., seasonal regime shifts, covariate shifts in demand patterns) that occur after the model has been deployed, making full retraining costly; **Low-complexity residual structure** means that residual errors often exhibit simple patterns (e.g., horizon-wise bias, scale miscalibration, calendar offsets) that can be captured by a small function class (tiny MLPs/low-rank heads) rather than requiring a new high-capacity backbone, but the base model fails to absorb them.

# B THEORETICAL PROOF

## B.1 PROOF OF PROPOSITION 2.1

Define risks (squared error):

$$\mathcal{R}_{\text{out}}(\delta) = \tfrac{1}{2}\,\mathbb{E}\big[\|Y - (F(X) + \delta g(X))\|^2\big] = \tfrac{1}{2}\,\mathbb{E}\big[\|R(X) - \delta g(X)\|^2\big], \tag{27}$$

and

$$\mathcal{R}_{\text{in}}(\delta) = \tfrac{1}{2}\,\mathbb{E}\big[\|Y - F(X + \delta u(X))\|^2\big]. \tag{28}$$

*Proof.* Let $A := \mathbb{E}\|g(X)\|^2$ and $B := \mathbb{E}\langle R(X), g(X)\rangle$. Then

$$\mathcal{R}_{\text{out}}(\delta) = \tfrac{1}{2}\,\mathbb{E}\|R\|^2 - \delta B + \tfrac{1}{2}\,\delta^2 A. \tag{29}$$

Hence $\mathcal{R}_{\text{out}}$ is a strictly convex quadratic in $\delta$ whenever $A > 0$, with unique minimizer $\delta^\star = B/A$ and minimal value

$$\mathcal{R}_{\text{out}}(\delta^\star) = \tfrac{1}{2}\,\mathbb{E}\|R\|^2 - \tfrac{1}{2}\,\frac{B^2}{A}. \tag{30}$$

In particular, if $B > 0$ and $A > 0$ then for all $0 < \delta < 2B/A$, $\mathcal{R}_{\text{out}}(\delta) < \mathcal{R}_{\text{out}}(0) = \tfrac{1}{2}\,\mathbb{E}\|R\|^2$. Then, expand the square:

$$\|R - \delta g\|^2 = \|R\|^2 - 2\delta\langle R, g\rangle + \delta^2\|g\|^2. \tag{31}$$

Taking expectations and multiplying by $\tfrac{1}{2}$ yields the displayed quadratic form. If $A > 0$, the derivative $d\mathcal{R}_{\text{out}}/d\delta = -B + \delta A$ vanishes uniquely at $\delta^\star = B/A$; strict convexity gives the minimal value above. If $B > 0$, then near $\delta = 0$ the derivative is negative, so every $\delta \in (0, 2B/A)$ strictly improves the risk over $\delta = 0$. If $A = 0$ then $g = 0$ a.s. and risk is constant; if $B \leq 0$ there is no positive $\delta$ improving over $\delta = 0$.

*Remark.* (i) This is exactly the first (shrunk) step of residual boosting. (ii) The achievable drop at the optimal $\delta^\star$ is $\tfrac{1}{2}(B^2/A)$, which is positive iff $B \neq 0$ and $A > 0$.

$\square$

### B.2 PROOF OF PROPOSITION 2.2

Let $F : \mathbb{R}^d \to \mathbb{R}^H$ be differentiable, $u : \mathcal{X} \to \mathbb{R}^d$ a measurable nudging field, and define for $\delta \geq 0$

$$\hat{Y}_{\text{in}}(X; \delta) = F\big(X + \delta\, u(X)\big), \qquad \mathcal{R}_{\text{in}}(\delta) = \tfrac{1}{2}\, \mathbb{E}\left[\big\| y - \hat{y}_{\text{in}}(X; \delta)\big\|_2^2\right]. \tag{32}$$

Write $r(X) = y - F(X)$, $J_F(X)$ for the Jacobian of $F$ at $X$, and $A := \mathbb{E}\big[\langle r(X),\, J_F(X)\, u(X)\rangle\big]$.

If $A > 0$, then there exists $\varepsilon > 0$ such that $\mathcal{R}_{\text{in}}(\delta) < \mathcal{R}_{\text{in}}(0)$ for all $\delta \in (0, \varepsilon]$. If, in addition, $F$ is affine in a neighborhood of the support of $X$ ($J_F$ is constant and the Hessian is zero), then

$$\mathcal{R}_{\text{in}}(\delta) = \tfrac{1}{2}\, \mathbb{E}\left[\| r(X) - \delta\, J_F\, u(X)\|_2^2\right], \tag{33}$$

is a quadratic function of $\delta$ whose unique minimizer is

$$\delta^\star = \frac{\mathbb{E}\left[\langle r(X),\, J_F\, u(X)\rangle\right]}{\mathbb{E}\left[\| J_F\, u(X)\|_2^2\right]}. \tag{34}$$

Based on the conditions, we know that: $F$ is $C^1$ (continuously differentiable) on an open set containing $\{x + \delta u(X) : \delta \in [0, \delta_0]\}$ for some $\delta_0 > 0$. And $\| J_F(X + \delta u(X))\, u(X)\|$ is integrable uniformly for $\delta \in [0, \delta_0]$, and $\| y - F(X + \delta u(X))\|$ is integrable.

First, we have the following lemma,

**Lemma 4** (Improvement via Jacobian-aligned nudging). *If $\mathbb{E}[\langle r, J_F u\rangle] > 0$, then sufficiently small $\delta > 0$ reduces risk. As before, the optimal small-step size is $\delta^\star = \frac{\mathbb{E}[\langle r, J_F u\rangle]}{\mathbb{E}[\| J_F u\|^2]}$.*

*Proof.* For each $X$, by the fundamental theorem of calculus in Banach spaces,

$$F\big(X + \delta u(X)\big) = F(X) + \int_0^\delta J_F\big(X + t\, u(X)\big)\, u(X)\, dt. \tag{35}$$

Hence,

$$R_\delta = R_0 - \int_0^\delta J_F\big(X + t\, u(X)\big)\, u(X)\, dt. \tag{36}$$

Let $F(\delta) := \mathcal{R}_{\text{in}}(\delta) = \tfrac{1}{2}\, \mathbb{E}\| R_\delta\|^2$. Using $\frac{d}{d\delta}\| v\|^2 = 2\langle v, v'\rangle$,

$$F'(\delta) = \mathbb{E}\big\langle R_\delta,\, -J_F\big(X + \delta u(X)\big)\, u(X)\big\rangle. \tag{37}$$

Under the domination assumption, dominated convergence allows $\delta \to 0$ inside the expectation, giving

$$F'(0) = -\, \mathbb{E}\langle R_0, J_F(X)u(X)\rangle. \tag{38}$$

If $C > 0$, then $F'(0) = -C < 0$. By continuity of $F'$ near 0 (again from dominated convergence and continuity of $J_F$), there exists $\varepsilon > 0$ so that $F$ is strictly decreasing on $(0, \varepsilon)$, hence $F(\delta) < F(0)$ for all $\delta \in (0, \varepsilon)$.

If, in addition, $\| J_F(z)\| \leq L_F$ and $\| u(X)\| \leq U(X)$ with $\mathbb{E}U(X)^2 < \infty$, then for $|\delta| \leq 1$, we have

$$\| F(X + \delta u) - F(X)\| \leq L_F\, |\delta|\, \| u(X)\|, \tag{39}$$

and the same quadratic expansion as in Proposition 2.1 yields

$$\mathcal{R}_{\text{in}}(\delta) \leq \tfrac{1}{2}\, \mathbb{E}\| R\|^2 - \delta\, \mathbb{E}\langle R, J_F u\rangle + \tfrac{1}{2}\, \delta^2\, L_F^2\, \mathbb{E}\| u\|^2, \tag{40}$$

making the "improvement for small $\delta$" explicit whenever $\mathbb{E}\langle R, J_F u\rangle > 0$. $\qquad \square$

#### B.2.1 PROOF OF STEP 1: EXACT SMALL-STEP DECREASE

*Proof.* Define, for each $(X, y)$,

$$f(\delta; x, y) := \tfrac{1}{2}\, \big\| y - F(X + \delta u(X))\big\|_2^2 = \tfrac{1}{2}\, \big\| r(\delta; x)\big\|_2^2, \quad r(\delta; x) := y - F(X + \delta u(X)). \tag{41}$$

By the chain rule,

$$\frac{\partial}{\partial \delta} f(\delta; x, y) = \left\langle r(\delta; x), \frac{\partial}{\partial \delta} r(\delta; x) \right\rangle = -\left\langle r(\delta; x), J_F\big(X + \delta u(X)\big) u(X) \right\rangle. \tag{42}$$

By the domination assumptions and Lemma 4, we can pass the derivative through the expectation to get

$$\mathcal{R}'_{\text{in}}(\delta) = \mathbb{E}\left[\frac{\partial}{\partial \delta} f(\delta; x, y)\right] = -\mathbb{E}\left[\left\langle r(\delta; x), J_F\big(X + \delta u(X)\big) u(X) \right\rangle\right]. \tag{43}$$

Evaluating at $\delta = 0$,

$$\mathcal{R}'_{\text{in}}(0) = -\mathbb{E}\left[\langle r(X), J_F(X) u(X) \rangle\right] = -A. \tag{44}$$

If $A > 0$, then $\mathcal{R}'_{\text{in}}(0) < 0$. By continuity of $\mathcal{R}'_{\text{in}}$ at 0, there exists $\varepsilon > 0$ such that $\mathcal{R}'_{\text{in}}(\delta) \leq -\frac{A}{2} < 0$ for all $\delta \in [0, \varepsilon]$. Therefore, for any $\delta \in (0, \varepsilon]$,

$$\mathcal{R}_{\text{in}}(\delta) - \mathcal{R}_{\text{in}}(0) = \int_0^\delta \mathcal{R}'_{\text{in}}(t)\, dt \leq -\frac{A}{2}\,\delta < 0, \tag{45}$$

which proves the strict risk decrease for sufficiently small positive $\delta$. $\qquad\square$

### B.2.2 Proof of step 2: closed-form $\delta^\star$ under an affine $F$

*Proof.* Assume $F$ is affine: $F(X) = Ax + b$ with a constant matrix $A \in \mathbb{R}^{H \times d}$. Then $J_F \equiv A$ and

$$F\big(X + \delta u(X)\big) = F(X) + \delta A u(X). \tag{46}$$

Hence,

$$\mathcal{R}_{\text{in}}(\delta) = \tfrac{1}{2}\mathbb{E}\left[\left\| r(X) - \delta\, A u(X) \right\|_2^2\right] = \tfrac{1}{2}\mathbb{E}\left[\|r(X)\|^2\right] - \delta\,\mathbb{E}\left[\langle r(X), A u(X) \rangle\right] + \tfrac{1}{2}\delta^2\,\mathbb{E}\left[\|A u(X)\|^2\right]. \tag{47}$$

This is a strictly convex quadratic in $\delta$ provided $\mathbb{E}[\|A u(X)\|^2] > 0$. Differentiating and setting to 0,

$$\mathcal{R}'_{\text{in}}(\delta) = -\mathbb{E}\left[\langle r(X), A u(X) \rangle\right] + \delta\,\mathbb{E}\left[\|A u(X)\|^2\right] = 0 \tag{48}$$

yields the unique minimizer

$$\delta^\star = \frac{\mathbb{E}\left[\langle r(X), A u(X) \rangle\right]}{\mathbb{E}\left[\|A u(X)\|^2\right]} = \frac{\mathbb{E}\left[\langle r(X), J_F u(X) \rangle\right]}{\mathbb{E}\left[\|J_F u(X)\|^2\right]}. \tag{49}$$

This completes the proof for affine $F$. The same expression arises if, instead of assuming affine $F$, we optimize the first-order surrogate obtained by linearizing $F$ at $\delta = 0$:

$$F\big(X + \delta u(X)\big) \approx F(X) + \delta\, J_F(X)\, u(X), \tag{50}$$

which leads to the quadratic proxy

$$\widetilde{\mathcal{R}}_{\text{in}}(\delta) := \tfrac{1}{2}\mathbb{E}\left[\left\| r(X) - \delta\, J_F(X) u(X) \right\|^2\right], \tag{51}$$

whose unique minimizer is the same $\delta^\star$ as above. Further, we denote by $H_F(X)[v, w] \in \mathbb{R}^H$ the second directional derivative of $F$ at $X$ along $v, w$, then

$$\mathcal{R}''_{\text{in}}(0) = \mathbb{E}\left[\|J_F(X) u(X)\|^2 - \langle r(X),\, H_F(X)[u(X), u(X)] \rangle\right]. \tag{52}$$

If there exists $\eta \in [0, 1)$ such that

$$\left|\mathbb{E}\left[\langle r(X),\, H_F(X)[u(X), u(X)] \rangle\right]\right| \leq \eta\,\mathbb{E}\left[\|J_F(X) u(X)\|^2\right], \tag{53}$$

then $\mathcal{R}''_{\text{in}}(0) \in [(1 - \eta)B_0,\, (1 + \eta)B_0]$ where $B_0 = \mathbb{E}\left[\|J_F u\|^2\right]$. In that case, the true local minimizer $\delta^\dagger$ of $\mathcal{R}_{\text{in}}$ satisfies the bracket

$$\frac{A}{(1 + \eta)B_0} \leq \delta^\dagger \leq \frac{A}{(1 - \eta)B_0}, \tag{54}$$

quantifying how curvature perturbs the first-order optimizer. When $F$ is affine or the curvature term averages to zero, $\eta = 0$ and $\delta^\dagger = \delta^\star$. $\qquad\square$

### B.3 PROOF OF PROPOSITION 3.1

*Proof.* By Lipschitzness of $F$:

$$\|\tilde{y} - \hat{y}\| = \|F(\tilde{X}) - F(X)\| \leq L_F \|\tilde{X} - X\| = L_F \, \delta \, \|A_\phi^{\text{in}}(X)\|. \tag{55}$$

According to $\|A_\phi^{\text{in}}(X)\|_\infty \leq 1$, $\|A_\phi^{\text{out}}(\hat{Y}, X)\|_\infty \leq 1$ and $\delta \in (0, \delta_{\max}]$ with $\delta_{\max} \leq 1$, we have $\|A_\phi^{\text{in}}(X)\| \leq \sqrt{Ld}\|A_\phi^{\text{in}}(X)\|_\infty \leq \sqrt{Ld}$. Combining yields the claim. □

### B.4 PROOF OF COROLLARY 1

*Proof.* Coordinatewise, $\tilde{x}_i - x_i = x_i\big(e^{\delta a_i} - 1\big)$. By the mean value theorem for $t \mapsto e^t$, for each $i$ there exists $\xi_i \in (0, \delta a_i)$ such that

$$e^{\delta a_i} - 1 = \delta a_i \, e^{\xi_i} \quad \Rightarrow \quad |\tilde{x}_i - x_i| = |x_i| \, \delta |a_i| \, e^{\xi_i} \leq B_X \, \delta |a_i| \, e^{|\xi_i|} \leq B_X \, \delta |a_i| \, e^{\delta \|a\|_\infty}. \tag{56}$$

According to $\|A_\phi^{\text{in}}(X)\|_\infty \leq 1$, $\|A_\phi^{\text{out}}(\hat{Y}, X)\|_\infty \leq 1$ and $\delta \in (0, \delta_{\max}]$ with $\delta_{\max} \leq 1$, we have $\|a\|_\infty \leq 1$, hence $e^{\delta \|a\|_\infty} \leq e^\delta$. Summing squares,

$$\|\tilde{x} - x\| = \sqrt{\sum_i |\tilde{x}_i - x_i|^2} \leq \sqrt{\sum_i (B_X \, \delta |a_i| \, e^\delta)^2} = \delta e^\delta B_X \|a\|. \tag{57}$$

Then apply Lipschitz step in Proposition 3.1, we have

$$\|\tilde{y} - \hat{y}\| = \|F(\tilde{X}) - F(X)\| \leq L_F \|\tilde{x} - x\| \leq \delta e^\delta L_F B_X \|a\|. \tag{58}$$

For $\delta \leq 1$, $e^\delta \leq e$, so the bound is $O(\delta)$. □

### B.5 PROOF OF THEOREM 2

*Proof.* By $\beta$-smoothness with $u = \tilde{y}$, $v = \hat{y}$,

$$\ell(\tilde{y}, y) \leq \ell(\hat{y}, y) + \nabla\ell(\hat{y}, y)^\top (\tilde{y} - \hat{y}) + \tfrac{\beta}{2}\|\tilde{y} - \hat{y}\|^2 = \ell(\hat{y}, y) + \delta\langle g, d\rangle + \tfrac{\beta}{2}\delta^2\|d\|^2. \tag{59}$$

By alignment condition, $\langle g, d\rangle \leq -\alpha\|g\|\|d\|$. Substitute:

$$\ell(\tilde{y}, y) - \ell(\hat{y}, y) \leq -\delta\alpha\|g\|\|d\| + \tfrac{\beta}{2}\delta^2\|d\|^2. \tag{60}$$

The RHS is a convex quadratic in $\delta$ with unique minimizer $\delta^\star = \frac{\alpha\|g\|}{\beta\|d\|}$. Plugging $\delta^\star$ gives $-\frac{\alpha^2}{2\beta}\|g\|^2$. Strict descent holds whenever the derivative at 0 is negative and the second-order term does not dominate, equivalently $\delta \in (0, \frac{2\alpha\|g\|}{\beta\|d\|})$. □

### B.6 PROOF OF THEOREM 3

*Proof.* By first-order Taylor expansion of $F$ at $x$,

$$F(x + \delta v) = F(X) + \delta Jv + r_F(\delta), \quad \text{with} \quad \|r_F(\delta)\| = O(\delta). \tag{61}$$

Set $\delta := \delta s + r_F(\delta)$, so $\tilde{y} = \hat{y} + \delta$. Apply $\beta$-smoothness of $\ell$:

$$\ell(\hat{y} + \delta, y) \leq \ell(\hat{y}, y) + \langle g, \delta\rangle + \tfrac{\beta}{2}\|\delta\|^2. \tag{62}$$

Then, compute the terms:

$$\langle g, \delta\rangle = \delta\langle g, s\rangle + \langle g, r_F(\delta)\rangle \quad \text{and} \quad \|\delta\|^2 = \delta^2\|s\|^2 + 2\delta\langle s, r_F(\delta)\rangle + \|r_F(\delta)\|^2. \tag{63}$$

Since $\|r_F(\delta)\| = O(\delta)$, we have $\langle g, r_F(\delta)\rangle = O(\delta)$ and $\|\delta\|^2 = \delta^2\|s\|^2 + O(\delta^2)$. Therefore,

$$\ell(F(x + \delta v), y) \leq \ell(\hat{y}, y) + \delta\langle g, s\rangle + \tfrac{\beta}{2}\delta^2\|s\|^2 + O(\delta^2). \tag{64}$$

If $\langle g, s\rangle \leq -\alpha\|g\|\|s\|$, then for sufficiently small $\delta$ the negative linear term dominates the $O(\delta^2)$ remainder, yielding strict descent. Optimizing the quadratic upper bound in $\delta$ gives the minimizer $\delta^\star = \frac{\alpha\|g\|}{\beta\|s\|}$ and value $-\frac{\alpha^2}{2\beta}\|g\|^2$ up to $O(1)$, establishing the last claim. □

### B.7 PROOF OF THEOREM 3.2

*Proof.* We formalize the two claims: (i) $O(\delta)$ bound on prediction drift, and (ii) loss upper bound under composition. Let the composed edit be: $\tilde{x} = x + \delta v$, $\hat{y}' := F(\tilde{X})$, and $\tilde{y} := \hat{y}' + \delta d(\hat{y}', X)$. As before, $\hat{y} = F(X)$.

(i) For $O(\delta)$ bound on prediction drift, using the triangle inequality, we have:

$$\|\tilde{y} - \hat{y}\| \leq \|\hat{y}' - \hat{y}\| + \delta\|d(\hat{y}', X)\|. \tag{65}$$

Further, according to $\|A_\phi^{\text{in}}(X)\|_\infty \leq 1$, $\|A_\phi^{\text{out}}(\hat{Y}, X)\|_\infty \leq 1$ and $\delta \in (0, \delta_{\max}]$ with $\delta_{\max} \leq 1$, we have $\|\hat{y}' - \hat{y}\| = \|F(\tilde{X}) - F(X)\| \leq L_F\|\tilde{x} - x\| = \delta L_F\|v\|$. The bound follows.

(ii) For loss upper bound under composition, by definition we have

$$\tilde{y} = \hat{y}' + \delta d' = \hat{y} + \delta s + r_F(\delta) + \delta d'. \tag{66}$$

Set $\Delta := \delta(s + d') + r_F(\delta)$. By $\beta$-smoothness, we have

$$\ell(\hat{y} + \Delta, y) \leq \ell(\hat{y}, y) + \langle g, \Delta \rangle + \tfrac{\beta}{2}\|\Delta\|^2, \tag{67}$$

which can be decomposed into:

$$\langle g, \Delta \rangle = \delta\langle g, s + d' \rangle + \langle g, r_F(\delta) \rangle, \tag{68}$$
$$\text{where} \quad \|\Delta\|^2 = \delta^2\|s + d'\|^2 + 2\delta\langle s + d', r_F(\delta) \rangle + \|r_F(\delta)\|^2. \tag{69}$$

Since $\|r_F(\delta)\| = O(\delta)$, we have $\langle g, r_F(\delta) \rangle = O(\delta)$ and $\|\Delta\|^2 = \delta^2\|s + d'\|^2 + O(\delta^2)$. Thus

$$\ell(\tilde{y}, y) \leq \ell(\hat{y}, y) + \delta\langle g, s + d' \rangle + \tfrac{\beta}{2}\delta^2\|s + d'\|^2 + O(\delta^2). \tag{70}$$

Here, if $\langle g, s + d' \rangle \leq -\alpha\|g\|\,\|s + d'\|$, the linear term is strictly negative whenever $s + d' \neq 0$. For sufficiently small $\delta$, the negative linear term dominates the $O(\delta^2)$ remainder, giving strict descent.

*Remark.* If a learned gate $\gamma \in [0, 1]^q$ combines input- and output-induced steps as $s_\gamma = \gamma \odot s + (1 - \gamma) \odot d'$, then alignment for $s_\gamma$ follows from mild conditions (e.g., selecting $\gamma$ to minimize $\langle g, s_\gamma \rangle$ subject to $\gamma \in [0, 1]^q$ ensures $\langle g, s_\gamma \rangle \leq \min\{\langle g, s \rangle, \langle g, d' \rangle\}$).

$\square$

## C EXPERIMENTAL SETUP AND RESULTS

### C.1 DATASET

### C.2 COMMONLY USED TS DATASETS

The information of the experiment datasets used in this paper are summarized as follows: (1) Electricity Transformer Temperature (ETT) dataset Zhou et al. (2021), which contains the data collected from two electricity transformers in two separated counties in China, including the load and the oil temperature recorded every 15 minutes (ETTm) or 1 hour (ETTh) between July 2016 and July 2018. (2) Electricity (ECL) dataset [1] collects the hourly electricity consumption of 321 clients (each column) from 2012 to 2014. (3) Exchange Lai et al. (2018) records the current exchange of 8 different countries from 1990 to 2016. (4) Traffic dataset [2] records the occupation rate of freeway system across State of California measured by 861 sensors. (5) Weather dataset [3] records every 10 minutes for 21 meteorological indicators in Germany throughout 2020. The detailed statistics information of the datasets is shown in Table 6.

---

[1] https://archive.ics.uci.edu/ml/datasets/ElectricityLoadDiagrams20112014
[2] http://pems.dot.ca.gov
[3] https://www.bgc-jena.mpg.de/wetter

Table 6: Details of the seven TS datasets.

| Dataset | length | features | frequency |
|---|---|---|---|
| ETTh1 | 17,420 | 7 | 1h |
| ETTh2 | 17,420 | 7 | 1h |
| ETTm1 | 69,680 | 7 | 15m |
| ETTm2 | 69,680 | 7 | 15m |
| Electricity | 26,304 | 321 | 1h |
| Exchange | 7,588 | 8 | 1d |
| Traffic | 17,544 | 862 | 1h |
| Weather | 52,696 | 21 | 10m |

### C.3 TRAINING OBJECTIVE

We train $\theta$ on $\mathcal{D}$ while backpropagating through $F$ but not updating it. Let $\tilde{Y}_\theta(X)$ denote the adapted prediction. For point forecasts we minimize a horizon-aware loss: Here are explicit formulas for each loss term when the input-adapter is a learnable mask $M(X;\phi) \in [0,1]^{L \times d}$ applied as $X' = X \odot M$. Let $\mathcal{D} = \{(X^{(i)}, Y^{(i)})\}_{i=1}^N$, $\hat{Y}^{(i)} = F(X^{(i)} \odot M(X^{(i)}; \phi))$, and $H, m$ be horizon and target dims. Expectations $\mathbb{E}$ below are over the empirical data distribution (mini-batches in practice).

MSE (point forecasts):

$$\mathcal{L}_{\text{pred}}^{\text{MSE}} = \mathbb{E}_{(X,Y) \sim \mathcal{D}} \left[ \frac{1}{Hm} \sum_{h=1}^{H} \sum_{k=1}^{m} w_h \left( \hat{Y}_{h,k} - Y_{h,k} \right)^2 \right]. \tag{71}$$

MAE (point forecasts):

$$\mathcal{L}_{\text{pred}}^{\text{MAE}} = \mathbb{E} \left[ \frac{1}{Hm} \sum_{h=1}^{H} \sum_{k=1}^{m} w_h \left| \hat{Y}_{h,k} - Y_{h,k} \right| \right]. \tag{72}$$

Pinball (quantile $\tau \in \mathcal{T}$). If $\hat{Y}^\tau$ predicts the $\tau$-quantile,

$$\mathcal{L}_{\text{pred}}^{\text{QB}} = \mathbb{E} \left[ \frac{1}{|\mathcal{T}|Hm} \sum_{\tau \in \mathcal{T}} \sum_{h,k} \rho_\tau \left( Y_{h,k} - \hat{Y}_{h,k}^\tau \right) \right], \quad \rho_\tau(u) = u\left( \tau - \mathbf{1}\{u < 0\} \right). \tag{73}$$

Sparsity (L1) on the mask:

$$\mathcal{L}_{\ell_1} = \mathbb{E}_{X \sim \mathcal{D}} \left[ \frac{1}{Ld} \sum_{t=1}^{L} \sum_{j=1}^{d} M_{t,j}(X; \phi) \right]. \tag{74}$$

Entropy (pushes mask toward 0 or 1):

$$\mathcal{L}_{\text{ent}} = \mathbb{E}_X \left[ -\frac{1}{Ld} \sum_{t,j} \left( M_{t,j} \log(M_{t,j} + \delta) + (1 - M_{t,j}) \log(1 - M_{t,j} + \delta) \right) \right]. \tag{75}$$

Temporal smoothness:

$$\mathcal{L}_{\text{TV}} = \mathbb{E}_X \left[ \frac{1}{(L-1)d} \sum_{t=2}^{L} \sum_{j=1}^{d} \left| M_{t,j}(X; \phi) - M_{t-1,j}(X; \phi) \right| \right]. \tag{76}$$

Budget (fraction of active entries not to exceed $\kappa$):

$$\bar{m}(X; \phi) = \frac{1}{Ld} \sum_{t,j} M_{t,j}(X; \phi). \tag{77}$$

A hinge penalty enforces $\bar{m} \leq \kappa$:

$$\mathcal{L}_{\text{bud}} = \mathbb{E}_X \left[ \left( \bar{m}(X; \phi) - \kappa \right)_+ \right], \qquad (u)_+ \equiv \max\{u, 0\}. \tag{78}$$

Group sparsity:

$$\mathcal{L}_{\text{group}} = \mathbb{E}_X \left[ \frac{1}{d} \sum_{j=1}^{d} \sqrt{\sum_{t=1}^{L} M_{t,j}(X; \phi)^2 + \delta} \right]. \tag{79}$$

## C.4 Online learning setup

During online testing, we set the batch size to 1 to ensure that data arrives in order. Meanwhile, we used a streaming buffer, where only one updated data point is cached at each moment/iteration (avoid label leakage raised by Liang et al. (2024a); yee Ava Lau et al. (2025), while returning a complete sample from a previous moment. E.g., at time $t$, the input used for online update returned from the buffer is $X_{t-H-L:t-H}$, the label is $X_{t-H:t}$, where $H$ is the prediction length and $L$ is the input length.

## C.5 Training Details of Ada-X+Y

Ada-X+Y is composed of Ada-X and Ada-Y, and Ada-X and Ada-Y are **trained jointly** in an end-to-end manner, not sequentially. We minimize a single combined loss $\mathcal{L}$ over the union of parameters $A_\theta^{in}$ (Ada-X) and $A_\theta^{out}$ (Ada-Y). The forward pass is:

$$\hat{Y} = F(X + \delta A_\theta^{in}(X)) \tag{80}$$
$$\tilde{Y} = \hat{Y} + \delta A_\theta^{out}(\hat{Y}) \tag{81}$$

During the backward pass, gradients flow from the loss through Ada-Y (Eq. 2), then through the backbone $F$, and finally to Ada-X (Eq. 1). This ensures that Ada-X learns input perturbations that specifically help the backbone produce features that Ada-Y can best correct.

Experimental Setup: In our experiments, we instantiate two separate Adam optimizers (both learning rate are 1E-4) for modular flexibility. However, they are stepped simultaneously after a single backward pass, making the process equivalent to optimizing a joint objective. As derived in Proposition 3.2, this joint update rule maintains the $O(\delta)$ drift bounds and descent guarantees, ensuring the two adapters do not destabilize each other.

## C.6 Using $\delta$-Adapters to Improve Multivariate Time Series

Tables 7, 8 and Figure 10 show that $\delta$-Adapter provides consistent improvements across multiple forecasting models. For nearly all datasets, Ada-X and Ada-Y lead to lower prediction errors compared to the original models, demonstrating that the proposed adapters generalize well to diverse forecasting architectures. Notably, Ada-X again delivers the largest gains, particularly on challenging datasets such as Exchange, Traffic, and ETT series, confirming that refining the input signals before model inference is the most impactful strategy. These results further validate that $\delta$-Adapter is a broadly applicable, efficient, and effective enhancement method for modern time series forecasting.

## C.7 $\delta$-Adapter's validation and testing performance changes with epochs

We also present the performance changes of additive and multiplicative adapters on different datasets over epochs (the blank is due to early stopping). In Figure 9, we visualize the changes in validation and test losses of $\delta$-Adapter across different datasets. Across Electricity, Traffic, and Weather, adding Ada-X+Y drives the test MSE consistently below the original frozen model from the very first epoch and then decreases further before plateauing after 5 epochs. Validation and test curves track closely (no divergence), indicating stable training without overfitting. The gains are monotonic

Table 7: Performances of the forecaster $F$ and $\delta$-Adapter under batch or online training.

| Dataset | Model | Original (Batch) | Fine-Turning (Batch) | Continue (Online) | Ada-X (Batch) | Ada-X (Online) | Ada-Y (Batch) | Ada-Y (Online) | Ada-X+Y (Batch) | Ada-X+Y (Online) |
|---|---|---|---|---|---|---|---|---|---|---|
| Weather | DistPred | 0.1710 | 0.1715 | 0.1700 | 0.1662 | 0.1654 | 0.1629 | 0.1623 | 0.1602 | 0.1560 |
| | iTransformer | 0.1731 | 0.1724 | 0.1721 | 0.1706 | 0.169 | 0.1631 | 0.1634 | 0.1609 | 0.1609 |
| Traffic | DistPred | 0.4229 | 0.4229 | 0.4229 | 0.4182 | 0.4179 | 0.4117 | 0.4119 | 0.4028 | 0.4029 |
| | iTransformer | 0.4437 | 0.4411 | 0.4414 | 0.4361 | 0.4360 | 0.4294 | 0.4294 | 0.4202 | 0.4202 |
| ELC | DistPred | 0.1546 | 0.1546 | 0.1545 | 0.1483 | 0.1476 | 0.1483 | 0.1474 | 0.1436 | 0.1432 |
| | iTransformer | 0.1655 | 0.1646 | 0.1636 | 0.1601 | 0.1596 | 0.1566 | 0.1562 | 0.1527 | 0.1515 |

Table 8: Multivariate time series forecasting results on the benchmark datasets.

| Dataset | | DistPred | | | iTransformer | | | FourierGNN | | | FreTS | | | Autoformer | | |
|---|---|---|---|---|---|---|---|---|---|---|---|---|---|---|---|---|
| | Length | Original | Ada-X | Ada-Y | Original | Ada-X | Ada-Y | Original | Ada-X | Ada-Y | Original | Ada-X | Ada-Y | Original | Ada-X | Ada-Y |
| ELC | 96 | 0.155 | 0.149 | 0.148 | 0.163 | 0.160 | 0.157 | 0.250 | 0.235 | 0.224 | 0.189 | 0.183 | 0.176 | 0.228 | 0.211 | 0.221 |
| | 192 | 0.169 | 0.166 | 0.162 | 0.175 | 0.173 | 0.167 | 0.255 | 0.245 | 0.230 | 0.193 | 0.189 | 0.180 | 0.437 | 0.383 | 0.384 |
| | 336 | 0.185 | 0.181 | 0.176 | 0.193 | 0.189 | 0.182 | 0.267 | 0.256 | 0.240 | 0.207 | 0.203 | 0.192 | 0.612 | 0.590 | 0.527 |
| | 720 | 0.221 | 0.217 | 0.190 | 0.231 | 0.226 | 0.218 | 0.298 | 0.284 | 0.268 | 0.246 | 0.239 | 0.229 | 0.782 | 0.767 | 0.670 |
| | Avg | 0.182 | 0.178 | 0.169 | 0.190 | 0.187 | 0.181 | 0.267 | 0.255 | 0.241 | 0.209 | 0.203 | 0.194 | 0.515 | 0.488 | 0.450 |
| ETTh1 | 96 | 0.389 | 0.385 | 0.384 | 0.390 | 0.385 | 0.386 | 0.506 | 0.502 | 0.503 | 0.397 | 0.398 | 0.396 | 0.449 | 0.437 | 0.444 |
| | 192 | 0.451 | 0.448 | 0.446 | 0.444 | 0.444 | 0.440 | 0.540 | 0.540 | 0.540 | 0.458 | 0.456 | 0.453 | 0.571 | 0.566 | 0.558 |
| | 336 | 0.498 | 0.493 | 0.497 | 0.479 | 0.478 | 0.483 | 0.583 | 0.585 | 0.584 | 0.507 | 0.508 | 0.511 | 0.656 | 0.644 | 0.638 |
| | 720 | 0.505 | 0.502 | 0.503 | 0.504 | 0.502 | 0.514 | 0.615 | 0.634 | 0.632 | 0.568 | 0.574 | 0.563 | 0.695 | 0.686 | 0.669 |
| | Avg | 0.461 | 0.457 | 0.458 | 0.454 | 0.453 | 0.456 | 0.561 | 0.566 | 0.565 | 0.482 | 0.484 | 0.481 | 0.593 | 0.583 | 0.577 |
| ETTh2 | 96 | 0.303 | 0.300 | 0.301 | 0.296 | 0.293 | 0.293 | 0.396 | 0.383 | 0.388 | 0.342 | 0.314 | 0.330 | 0.375 | 0.358 | 0.375 |
| | 192 | 0.378 | 0.372 | 0.373 | 0.383 | 0.380 | 0.380 | 0.507 | 0.468 | 0.477 | 0.468 | 0.427 | 0.437 | 0.438 | 0.432 | 0.444 |
| | 336 | 0.447 | 0.438 | 0.443 | 0.429 | 0.427 | 0.434 | 0.558 | 0.500 | 0.500 | 0.548 | 0.501 | 0.506 | 0.464 | 0.460 | 0.459 |
| | 720 | 0.431 | 0.433 | 0.431 | 0.445 | 0.440 | 0.453 | 0.718 | 0.646 | 0.660 | 0.791 | 0.725 | 0.717 | 0.473 | 0.470 | 0.494 |
| | Avg | 0.390 | 0.386 | 0.387 | 0.388 | 0.385 | 0.390 | 0.545 | 0.499 | 0.506 | 0.537 | 0.492 | 0.498 | 0.438 | 0.420 | 0.423 |
| ETTm1 | 96 | 0.339 | 0.324 | 0.330 | 0.345 | 0.334 | 0.334 | 0.405 | 0.399 | 0.397 | 0.340 | 0.334 | 0.337 | 0.586 | 0.464 | 0.569 |
| | 192 | 0.384 | 0.374 | 0.378 | 0.382 | 0.369 | 0.374 | 0.435 | 0.427 | 0.430 | 0.380 | 0.378 | 0.380 | 0.627 | 0.572 | 0.602 |
| | 336 | 0.416 | 0.409 | 0.404 | 0.431 | 0.423 | 0.419 | 0.464 | 0.457 | 0.455 | 0.417 | 0.414 | 0.415 | 0.691 | 0.650 | 0.656 |
| | 720 | 0.510 | 0.487 | 0.495 | 0.511 | 0.501 | 0.496 | 0.519 | 0.506 | 0.507 | 0.483 | 0.478 | 0.474 | 0.754 | 0.729 | 0.720 |
| | Avg | 0.412 | 0.399 | 0.402 | 0.417 | 0.407 | 0.406 | 0.456 | 0.447 | 0.447 | 0.405 | 0.401 | 0.401 | 0.664 | 0.604 | 0.637 |
| ETTm2 | 96 | 0.179 | 0.175 | 0.179 | 0.182 | 0.179 | 0.183 | 0.220 | 0.204 | 0.213 | 0.191 | 0.179 | 0.187 | 0.271 | 0.236 | 0.288 |
| | 192 | 0.245 | 0.242 | 0.245 | 0.255 | 0.251 | 0.253 | 0.329 | 0.289 | 0.322 | 0.275 | 0.241 | 0.270 | 0.290 | 0.286 | 0.289 |
| | 336 | 0.309 | 0.304 | 0.302 | 0.327 | 0.317 | 0.323 | 0.380 | 0.359 | 0.380 | 0.342 | 0.309 | 0.333 | 0.359 | 0.350 | 0.350 |
| | 720 | 0.406 | 0.394 | 0.404 | 0.435 | 0.423 | 0.414 | 0.852 | 0.694 | 0.842 | 0.531 | 0.413 | 0.501 | 0.435 | 0.431 | 0.432 |
| | Avg | 0.285 | 0.279 | 0.282 | 0.300 | 0.292 | 0.293 | 0.445 | 0.386 | 0.439 | 0.335 | 0.285 | 0.323 | 0.339 | 0.316 | 0.320 |
| Exchange | 96 | 0.084 | 0.088 | 0.084 | 0.099 | 0.102 | 0.099 | 0.106 | 0.118 | 0.105 | 0.105 | 0.098 | 0.105 | 0.195 | 0.170 | 0.179 |
| | 192 | 0.190 | 0.186 | 0.182 | 0.180 | 0.173 | 0.168 | 0.208 | 0.216 | 0.203 | 0.186 | 0.181 | 0.187 | 0.260 | 0.243 | 0.246 |
| | 336 | 0.319 | 0.281 | 0.294 | 0.352 | 0.304 | 0.329 | 0.365 | 0.396 | 0.368 | 0.383 | 0.380 | 0.386 | 0.437 | 0.414 | 0.402 |
| | 720 | 0.809 | 0.653 | 0.714 | 0.901 | 0.814 | 0.800 | 0.841 | 0.843 | 0.841 | 0.989 | 0.989 | 1.011 | 1.144 | 1.095 | 1.021 |
| | Avg | 0.350 | 0.302 | 0.319 | 0.383 | 0.348 | 0.349 | 0.380 | 0.393 | 0.379 | 0.416 | 0.412 | 0.422 | 0.509 | 0.481 | 0.462 |
| Traffic | 96 | 0.423 | 0.416 | 0.412 | 0.444 | 0.436 | 0.429 | 0.779 | 0.753 | 0.731 | 0.563 | 0.555 | 0.538 | 0.659 | 0.657 | 0.650 |
| | 192 | 0.441 | 0.435 | 0.429 | 0.460 | 0.455 | 0.446 | 0.756 | 0.721 | 0.710 | 0.568 | 0.562 | 0.545 | 0.829 | 0.814 | 0.804 |
| | 336 | 0.458 | 0.453 | 0.447 | 0.479 | 0.475 | 0.465 | 0.765 | 0.739 | 0.739 | 0.595 | 0.589 | 0.572 | 1.094 | 1.072 | 1.025 |
| | 720 | 0.490 | 0.487 | 0.480 | 0.517 | 0.513 | 0.502 | 0.806 | 0.781 | 0.780 | 0.659 | 0.653 | 0.634 | 1.307 | 1.292 | 1.195 |
| | Avg | 0.453 | 0.448 | 0.442 | 0.475 | 0.470 | 0.461 | 0.777 | 0.749 | 0.740 | 0.596 | 0.590 | 0.572 | 0.972 | 0.959 | 0.918 |
| Weather | 96 | 0.171 | 0.166 | 0.162 | 0.173 | 0.165 | 0.163 | 0.184 | 0.183 | 0.172 | 0.185 | 0.177 | 0.172 | 0.262 | 0.240 | 0.243 |
| | 192 | 0.224 | 0.220 | 0.213 | 0.223 | 0.213 | 0.210 | 0.226 | 0.222 | 0.214 | 0.224 | 0.218 | 0.212 | :0.3003 | 0.282 | 0.275 |
| | 336 | 0.278 | 0.270 | 0.264 | 0.284 | 0.271 | 0.267 | 0.273 | 0.268 | 0.260 | 0.272 | 0.267 | 0.260 | 0.323 | 0.315 | 0.312 |
| | 720 | 0.353 | 0.347 | 0.340 | 0.357 | 0.349 | 0.339 | 0.338 | 0.330 | 0.329 | 0.341 | 0.335 | 0.328 | 0.389 | 0.385 | 0.367 |
| | Avg | 0.256 | 0.251 | 0.245 | 0.259 | 0.249 | 0.245 | 0.255 | 0.251 | 0.244 | 0.255 | 0.249 | 0.243 | 0.325 | 0.306 | 0.299 |

or near-monotonic on Electricity and Traffic, while Weather shows an immediate, steady improvement that remains well under the original baseline. Overall, Ada-X+Y delivers fast convergence and robust generalization across datasets. These experiments show that the loss curve of the $\delta$-Adapter gradually decreases with epochs and has stable and consistent boundaries. Meanwhile, the composite adapter (X+Y) can achieve better performance (Stability Analysis of Section 3), which also proves the robustness of the $\delta$-Adapter and the correctness of its theoretical foundation.

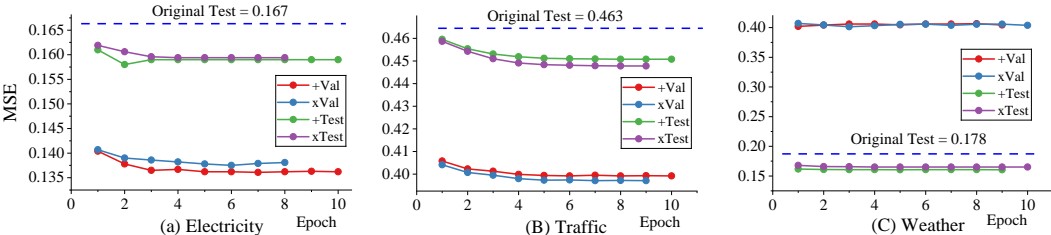

Figure 9: Validation and testing performance changes with epochs when adding $\delta$-Adapter.

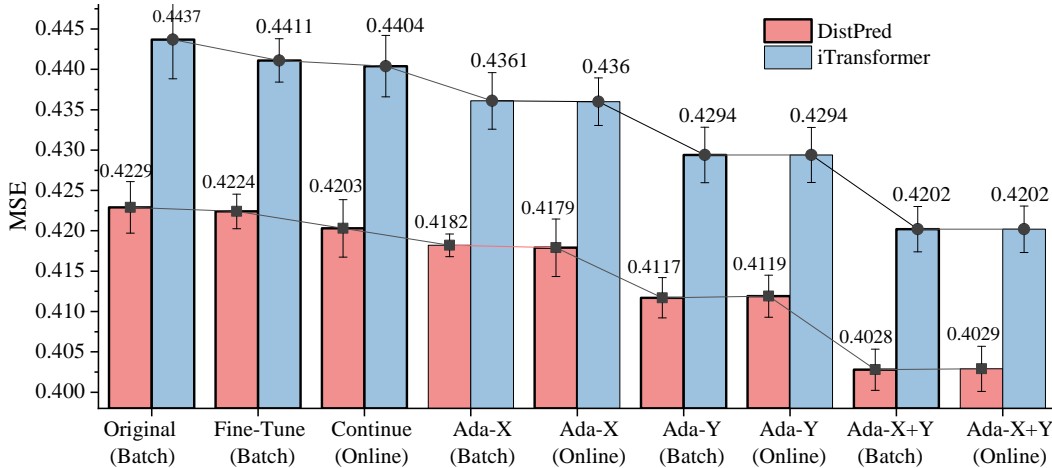

Figure 10: Performances of $\delta$-Adapter under batch or online training

## C.8 EFFICIENCY OF $\delta$-ADAPTER

We tested the efficiency of $\delta$-Adapter. Table 9 shows that $\delta$-Adapter is the most time-efficient adaptive method overall, it is consistently faster than other methods across all horizons. This is because Ada-X+Y itself is lightweight and only uses the most recent single sample to update the model. Compared to other adapters that use or select a large number of recent samples for updates, it is obviously faster. The $\delta$-Adapter is designed to be extremely lightweight. Compared to the backbone model (specifically, for Sundial (128M) and TabPFN (48M), the adapter introduces less than 2%-6% additional parameters, validating the lightweight claim.

Table 9: Time (S) and memory (MB) of adapters (backbone is TabPFN) and online methods.

| TabPFN 48M | | Ada-X+Y 3M | | | SOLID 0.5M | | | TAFAS 6M | | | OneNet 3M | | | FSNet 2M | | |
|---|---|---|---|---|---|---|---|---|---|---|---|---|---|---|---|---|
| Time | Memory | Train | Test | Memory | Train | Test | Memory | Train | Test | Memory | Train | Time | Memory | Train | Time | Memory |
| 281 | 1840 | 392 | 395 | 1983 | 511 | 667 | 2401 | 603 | 861 | 3468 | 693 | 471 | 1512 | 621 | 485 | 1504 |
| 307 | 1848 | 386 | 379 | 2132 | 481 | 624 | 2423 | 589 | 895 | 3790 | 681 | 445 | 1537 | 618 | 472 | 1531 |
| 326 | 1852 | 385 | 415 | 2622 | 484 | 593 | 2446 | 583 | 1152 | 4186 | 631 | 452 | 1559 | 599 | 466 | 1517 |
| 351 | 1856 | 369 | 431 | 3102 | 505 | 398 | 2501 | 916 | 1803 | 6809 | 530 | 465 | 1567 | 554 | 458 | 1526 |

## C.9 QUANTILE CALIBRATOR (QC) AND CONFORMAL CALIBRATOR (CC)

Figure 11 illustrates that both calibrators produce well-calibrated intervals. QC attains higher coverage than CC, while on another sample CC is better. QC tends to yield slightly wider, more conservative bands. CC delivers comparably high coverage with tighter intervals. Overall, the two methods are complementary and reliably improve uncertainty quantification over the raw predictor.

Now, let's discuss how to choose between QC and CC. Both modules turn a frozen point forecaster into a calibrated probabilistic predictor, but they are aimed at slightly different desiderata: QC directly learns horizon-wise conditional quantiles as bounded offsets around the point forecast, which produces a smooth quantile function over multiple levels without assumptions about the underlying distribution. CC learns only a heteroscedastic scale function and combines it with normalized-residual conformal prediction on a held-out calibration set, yielding symmetric but input-dependent intervals with finite-sample marginal coverage under exchangeability.

Empirically, both variants achieve strong coverage, but QC tends to produce marginally wider and more conservative bands, while CC attains similar coverage with somewhat tighter intervals (see Figs. 5 and 6). For a new real-world dataset, our recommendation is therefore: If strict coverage guarantees are the main requirement, CC is preferable, because the conformal step provides finite-sample marginal coverage at the target level. If one needs a rich predictive distribution or multiple coverage levels from a single model, QC is more convenient, as it directly returns a full quantile curve while remaining non-parametric w.r.t. the underlying distribution.

## C.10 COMPARISON BETWEEN ADAPTERS AND ONLINE LEARNING METHODS

The adapter-based methods we reviewed include SOLID, TAFAS, etc., and online approaches, e.g., FSNet and OneNet. These methods aim to mitigate test-time concept drift via selective layer retraining (SOLID), online adapter updates (TAFAS), auxiliary loss (PETSA), layer-wise adjustment and memory (FSNet), and dynamic model selection (OneNet). However, according to works by Liang et al. (2024a); yee Ava Lau et al. (2025), the above methods have used future labels to some extent, causing label leakage in long-term forecasting, where future ground truth is adopted in advance for adaptation. To achieve a fair comparison, we removed label leakage (it may cause performance degradation of some methods) to test their performance. As shown in Table 10, it can be found that our method achieves the lowest error on every dataset across all backbones.

Table 10: Comparison of various adapters and online methods.

| Model | | DistPred | | | | iTransformer | | | | Autoformer | | | | Others | |
|---|---|---|---|---|---|---|---|---|---|---|---|---|---|---|---|---|
| Dataset | Length | Offline | SOLID | TAFAS | Ada-X+Y | Offline | SOLID | TAFAS | Ada-X+Y | Offline | SOLID | TAFAS | Ada-X+Y | OneNet† | FSNet† |
| ELC | 96 | 0.155 | 0.154 | 0.156 | 0.146 | 0.163 | 0.165 | 0.165 | 0.156 | 0.228 | 0.211 | 0.230 | 0.196 | 0.247 | 0.310 |
| | 192 | 0.169 | 0.170 | 0.170 | 0.164 | 0.175 | 0.177 | 0.176 | 0.167 | 0.437 | 0.433 | 0.442 | 0.379 | 0.300 | 0.442 |
| | 336 | 0.185 | 0.182 | 0.183 | 0.178 | 0.193 | 0.189 | 0.189 | 0.176 | 0.612 | 0.583 | 0.588 | 0.580 | 0.325 | 0.483 |
| | 720 | 0.221 | 0.220 | 0.220 | 0.213 | 0.231 | 0.230 | 0.231 | 0.222 | 0.782 | 0.780 | 0.781 | 0.757 | 0.798 | 0.913 |
| | Avg | 0.182 | 0.182 | 0.182 | **0.175** | 0.190 | 0.190 | 0.190 | **0.180** | 0.515 | 0.502 | 0.510 | **0.478** | 0.417 | 0.537 |
| ETTh1 | 96 | 0.389 | 0.392 | 0.393 | 0.381 | 0.390 | 0.391 | 0.394 | 0.382 | 0.449 | 0.444 | 0.442 | 0.431 | 0.524 | 0.730 |
| | 192 | 0.451 | 0.450 | 0.452 | 0.445 | 0.444 | 0.450 | 0.437 | 0.440 | 0.571 | 0.566 | 0.565 | 0.561 | 0.571 | 0.820 |
| | 336 | 0.498 | 0.495 | 0.534 | 0.480 | 0.479 | 0.481 | 0.512 | 0.474 | 0.656 | 0.652 | 0.653 | 0.635 | 0.614 | 0.899 |
| | 720 | 0.505 | 0.502 | 0.524 | 0.497 | 0.504 | 0.509 | 0.563 | 0.499 | 0.695 | 0.694 | 0.705 | 0.681 | 0.762 | 1.060 |
| | Avg | 0.461 | 0.460 | 0.476 | **0.451** | 0.454 | 0.458 | 0.477 | **0.449** | 0.593 | 0.589 | 0.591 | **0.577** | 0.618 | 0.877 |
| ETTh2 | 96 | 0.303 | 0.320 | 0.319 | 0.294 | 0.296 | 0.311 | 0.311 | 0.286 | 0.375 | 0.371 | 0.381 | 0.353 | 0.515 | 0.515 |
| | 192 | 0.378 | 0.371 | 0.413 | 0.364 | 0.383 | 0.385 | 0.412 | 0.375 | 0.438 | 0.439 | 0.437 | 0.430 | 0.568 | 0.572 |
| | 336 | 0.447 | 0.445 | 0.447 | 0.432 | 0.429 | 0.429 | 0.498 | 0.423 | 0.464 | 0.453 | 0.452 | 0.456 | 0.602 | 0.615 |
| | 720 | 0.431 | 0.428 | 0.504 | 0.427 | 0.445 | 0.446 | 0.570 | 0.425 | 0.473 | 0.479 | 0.475 | 0.465 | 0.637 | 0.646 |
| | Avg | 0.390 | 0.391 | 0.402 | **0.379** | 0.388 | 0.393 | 0.448 | **0.377** | 0.438 | 0.435 | 0.436 | **0.426** | 0.581 | 0.587 |
| ETTm1 | 96 | 0.339 | 0.340 | 0.339 | 0.318 | 0.345 | 0.341 | 0.345 | 0.331 | 0.586 | 0.588 | 0.512 | 0.461 | 0.435 | 0.655 |
| | 192 | 0.384 | 0.389 | 0.387 | 0.373 | 0.382 | 0.381 | 0.384 | 0.362 | 0.627 | 0.628 | 0.642 | 0.564 | 0.496 | 0.825 |
| | 336 | 0.416 | 0.412 | 0.414 | 0.408 | 0.431 | 0.423 | 0.437 | 0.420 | 0.691 | 0.689 | 0.673 | 0.642 | 0.585 | 0.867 |
| | 720 | 0.510 | 0.484 | 0.504 | 0.484 | 0.511 | 0.510 | 0.512 | 0.500 | 0.754 | 0.740 | 0.725 | 0.721 | 0.676 | 1.055 |
| | Avg | 0.412 | 0.406 | 0.411 | **0.396** | 0.417 | 0.414 | 0.420 | **0.403** | 0.664 | 0.661 | 0.638 | **0.597** | 0.548 | 0.851 |
| ETTm2 | 96 | 0.179 | 0.179 | 0.180 | 0.171 | 0.182 | 0.182 | 0.183 | 0.176 | 0.271 | 0.267 | 0.273 | 0.233 | 0.434 | 0.334 |
| | 192 | 0.245 | 0.247 | 0.249 | 0.237 | 0.255 | 0.254 | 0.259 | 0.249 | 0.290 | 0.298 | 0.295 | 0.281 | 0.602 | 0.873 |
| | 336 | 0.309 | 0.311 | 0.311 | 0.298 | 0.327 | 0.325 | 0.332 | 0.315 | 0.359 | 0.357 | 0.353 | 0.348 | 0.829 | 1.156 |
| | 720 | 0.406 | 0.402 | 0.411 | 0.391 | 0.435 | 0.432 | 0.440 | 0.420 | 0.435 | 0.432 | 0.431 | 0.423 | 2.819 | 2.090 |
| | Avg | 0.285 | 0.285 | 0.288 | **0.274** | 0.300 | 0.298 | 0.304 | **0.290** | 0.339 | 0.339 | 0.338 | **0.321** | 1.171 | 1.113 |
| Exchange | 96 | 0.084 | 0.081 | 0.080 | 0.085 | 0.099 | 0.097 | 0.098 | 0.082 | 0.195 | 0.166 | 0.175 | 0.165 | 0.338 | 0.709 |
| | 192 | 0.190 | 0.189 | 0.189 | 0.182 | 0.180 | 0.177 | 0.176 | 0.167 | 0.260 | 0.234 | 0.237 | 0.236 | 0.591 | 0.771 |
| | 336 | 0.319 | 0.313 | 0.374 | 0.279 | 0.352 | 0.359 | 0.420 | 0.300 | 0.437 | 0.434 | 0.432 | 0.407 | 0.617 | 0.848 |
| | 720 | 0.809 | 0.806 | 0.808 | 0.642 | 0.901 | 0.870 | 0.873 | 0.714 | 1.144 | 1.130 | 1.134 | 1.050 | 1.041 | 1.183 |
| | Avg | 0.350 | 0.347 | 0.363 | **0.297** | 0.383 | 0.376 | 0.392 | **0.316** | 0.509 | 0.491 | 0.495 | **0.465** | 0.647 | 0.878 |
| Traffic | 96 | 0.423 | 0.424 | 0.424 | 0.410 | 0.444 | 0.445 | 0.443 | 0.426 | 0.659 | 0.634 | 0.664 | 0.653 | 0.546 | 0.677 |
| | 192 | 0.441 | 0.447 | 0.447 | 0.431 | 0.460 | 0.463 | 0.467 | 0.448 | 0.829 | 0.827 | 0.844 | 0.804 | 0.549 | 0.690 |
| | 336 | 0.458 | 0.451 | 0.457 | 0.443 | 0.479 | 0.475 | 0.472 | 0.456 | 1.094 | 1.080 | 1.096 | 1.030 | 0.571 | 0.705 |
| | 720 | 0.490 | 0.490 | 0.493 | 0.475 | 0.517 | 0.515 | 0.523 | 0.513 | 1.307 | 1.296 | 1.297 | 1.282 | 0.603 | 0.732 |
| | Avg | 0.453 | 0.453 | 0.455 | **0.440** | 0.475 | 0.475 | 0.476 | **0.461** | 0.972 | 0.959 | 0.975 | **0.942** | 0.567 | 0.701 |
| Weather | 96 | 0.171 | 0.168 | 0.170 | 0.160 | 0.173 | 0.170 | 0.172 | 0.162 | 0.262 | 0.260 | 0.299 | 0.235 | 0.251 | 0.322 |
| | 192 | 0.224 | 0.224 | 0.226 | 0.211 | 0.223 | 0.221 | 0.222 | 0.210 | :0.3003 | 0.298 | 0.295 | 0.276 | 0.295 | 0.465 |
| | 336 | 0.278 | 0.275 | 0.276 | 0.254 | 0.284 | 0.281 | 0.282 | 0.261 | 0.323 | 0.321 | 0.321 | 0.309 | 0.316 | 0.514 |
| | 720 | 0.353 | 0.353 | 0.353 | 0.342 | 0.357 | 0.357 | 0.358 | 0.345 | 0.389 | 0.386 | 0.384 | 0.375 | 0.697 | 0.862 |
| | Avg | 0.256 | 0.255 | 0.256 | **0.242** | 0.259 | 0.257 | 0.259 | **0.244** | 0.325 | 0.316 | 0.325 | **0.299** | 0.390 | 0.541 |

† OneNet and FSNet are implemented based on the public library provided in this paper, with their backbone models derived from their respective literatures. This implementation removes concept drift, and as a result, the online learning performance has deteriorated.

## C.11 ABLATION STUDIES OF δ-ADAPTER'S DEPTH, WIDTH AND VALUE

Table 11 shows that ablation studies of $\delta$-Adapter's depth and width. It can be found that the depth has little impact on performance, while the greater the width, the slight improvement in performance.

Table 11: Ablation studies of $\delta$-Adapter's depth and width.

| Depth | | 2 | | | | 3 | | | | 4 | | |
|---|---|---|---|---|---|---|---|---|---|---|---|---|
| Width | 64 | 128 | 256 | 512 | 64 | 128 | 256 | 512 | 64 | 128 | 256 | 512 |
| ELC | 0.159 | 0.157 | 0.155 | 0.1533 | 0.158 | 0.157 | 0.154 | 0.152 | 0.159 | 0.157 | 0.154 | 0.152 |
| Weather | 0.162 | 0.16 | 0.159 | 0.158 | 0.162 | 0.161 | 0.159 | 0.158 | 0.162 | 0.161 | 0.16 | 0.158 |
| Traffic | 0.439 | 0.437 | 0.433 | 0.43 | 0.44 | 0.436 | 0.433 | 0.43 | 0.44 | 0.436 | 0.433 | 0.43 |

## C.12 THE CHOICE OF HYPERPARAMETER δ

$\delta$ is related to the properties of the dataset (e.g., noise level, degree of concept drift). In our work, we divided the datasets into two categories: one with severe concept drift ($\delta = 0.1$, e.g., Traffic,

Weather, etc.) and the other with non-severe concept drift ($\delta = 0.01$, e.g., ETT, etc.). We did not perform hyperparameter searches based on models or datasets; instead, for datasets with severe concept drift, setting $\delta = 0.1$ is sufficient. In addition, we conducted ablation experiments on $\delta$. As shown in Table 12, a better value of $\delta = 0.1$ might yield better results. In our paper, we only reported the two settings ($\delta$=0.1 or 0.01).

Table 12: Ablation studies of $\delta$-Adapter's value.

| +0.1X | | ×0.1X | | ×0.2X | | +0.1Y | | ×0.1Y | | ×0.2Y | | +0.1(X&Y) | | ×0.1(X&Y) | | ×0.2(X&Y) | |
|---|---|---|---|---|---|---|---|---|---|---|---|---|---|---|---|---|---|
| Val | Test | Val | Test | Val | Test | Val | Test | Val | Test | Val | Test | Val | Test | Val | Test | Val | Test |
| 0.418 | 0.166 | 0.427 | 0.168 | 0.425 | 0.169 | 0.421 | 0.168 | 0.415 | 0.165 | 0.413 | 0.167 | 0.413 | 0.160 | 0.416 | 0.162 | 0.411 | 0.162 |

### C.13    PERFORMANCE OF $\delta$-ADAPTER ON BLACK-BOX MODELS.

Table 13 shows the performance of the $\delta$-Adapter on the black-box models. Specifically, we used TabPFN Hollmann et al. (2025) and TimesFM Das et al. (2024) as frozen black-box models to conduct zero-shot testing on various datasets. For comparison, we corrected the output results of these black-box models by adding Ada-Y. As shown in the table below, it can be found that after adding Ada-Y, the prediction error of the model is significantly reduced, which further proves the effectiveness of the proposed method.

### C.14    VISUALIZATION OF FEATURE SELECTOR AND CORRECTOR

Figure 11 shows that both Quantile Calibrator (QC) and Conformal Calibrator (CC) produce adaptive, heteroscedastic prediction intervals that track signal volatility—widening near peaks/troughs and typically enclosing the ground truth across diverse samples. QC tends to be more conservative (wider bands) and sometimes attains higher per-sample coverage (e.g., PICP≈0.729), while CC achieves tighter intervals with comparable coverage (e.g., PICP≈0.677 on multiple samples). The consistent behavior across the two test sets indicates that the calibrators are complementary and robust, yielding reliable uncertainty quantification beyond the raw predictor.

Figure 12 present the visualization results of the feature selector. We selected 90%, 50%, 30%, and 10% of the input data respectively to test the pre-trained iTransformer model, in order to observe their impact on the output results. It can be seen from the figure that most of the important features selected by the $\delta$-Adapter determine the performance of the model, while other features are relatively less important.

Table 13: Performance of $\delta$-Adapter on black-box models.

|  | Traffic | Weather | ELC | Exchange | ETTh1 | ETTh2 | ETTm1 | ETTm2 |
|---|---|---|---|---|---|---|---|---|
| TabPFN | 0.367 | 0.875 | 0.115 | 0.129 | 0.129 | 0.180 | 0.037 | 0.114 |
| Ada-Y | **0.342** | **0.552** | **0.089** | **0.096** | **0.095** | **0.171** | **0.034** | **0.103** |
| TimesFM | 0.211 | 0.168 | 0.084 | 0.239 | 0.029 | 0.135 | 0.028 | 0.267 |
| Ada-Y | **0.196** | **0.157** | **0.081** | **0.215** | **0.024** | **0.104** | **0.025** | **0.223** |

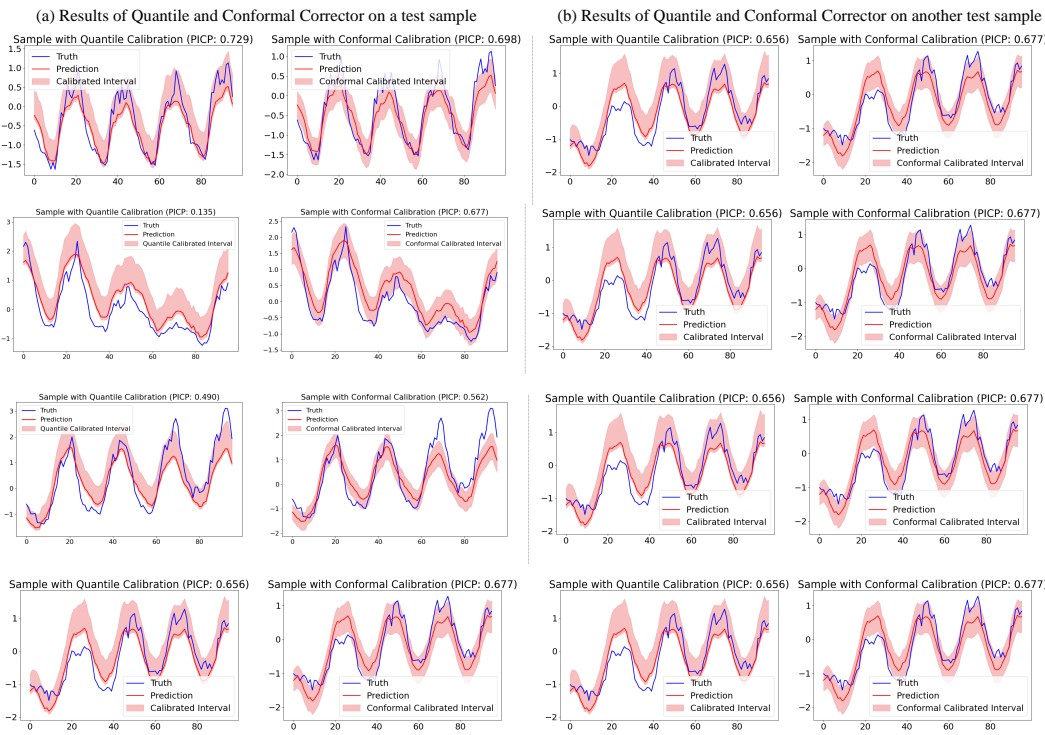

Figure 11: Visualization of the Quantile and Conformal calibrator predictions.

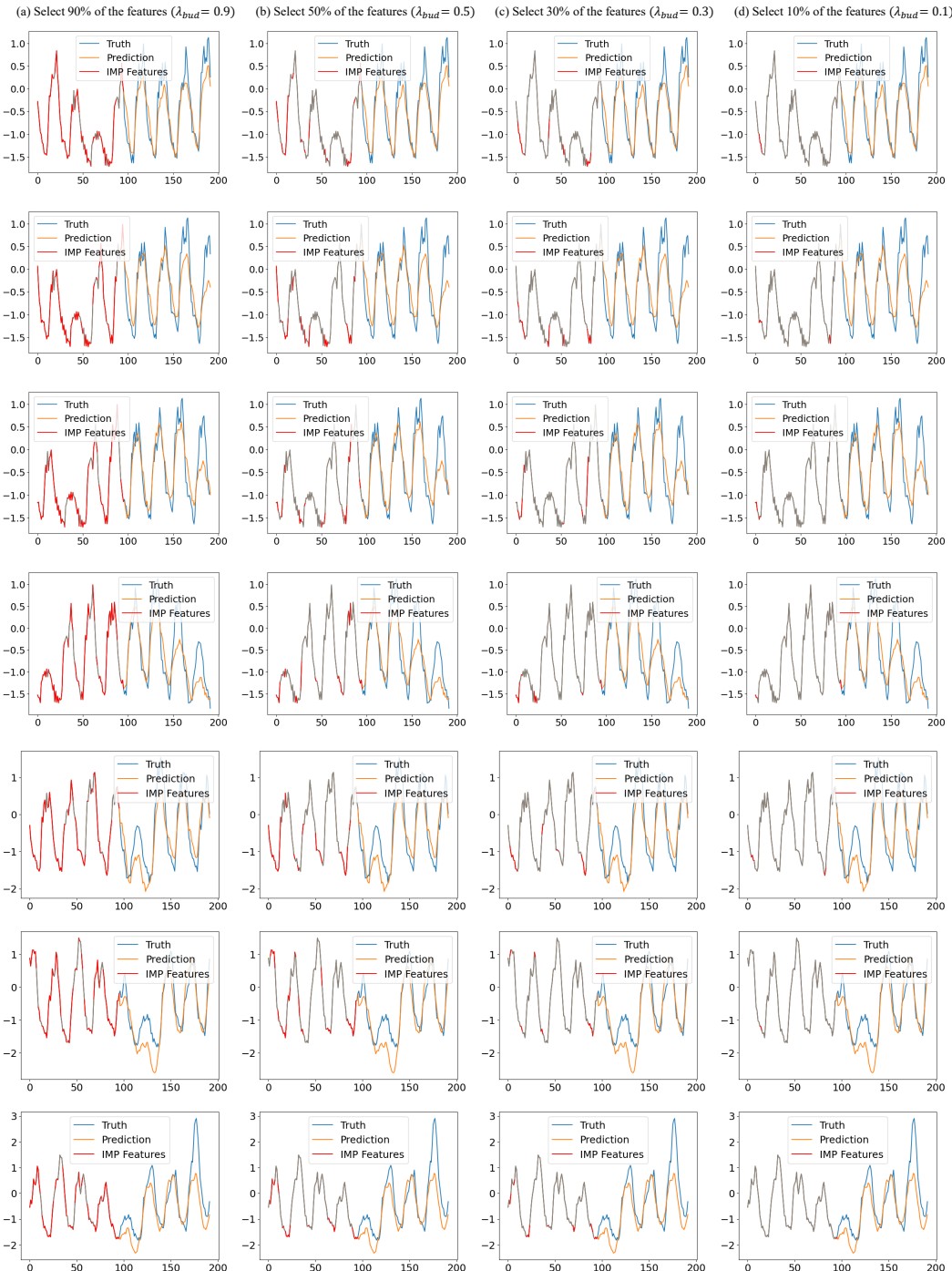

Figure 12: Visualization of different important features learned by the mask adapter.

