# OpenReview forum: "The Forecast After the Forecast: A Post-Processing Shift in Time Series"
_ICLR.cc/2026/Conference — ICLR 2026 Poster_

### Official Review · Reviewer_Fzqc · 2025-10-27

**Soundness:** 3
**Presentation:** 3
**Contribution:** 3
**Rating:** 6
**Confidence:** 4

**Summary:**

The paper proposes **δ-Adapter**, a lightweight and architecture-agnostic framework that improves frozen time-series forecasting models through post-processing rather than retraining. It introduces two modules: **input nudging**, which softly edits covariates before inference, and **output residual correction**, which refines predictions after inference. The method provides theoretical guarantees of local descent, bounded drift, and compositional stability. Additionally, the authors design a **feature-selector adapter** that identifies influential input features and **quantile/conformal calibrators** that enhance uncertainty estimation. Experiments across multiple forecasting backbones and datasets demonstrate that δ-Adapter consistently improves accuracy and calibration with minimal computational cost.

**Strengths:**

The paper’s main strength lies in its original and practical reformulation of time-series improvement as a post-processing problem rather than a model-design or retraining task. This shift in perspective is both creative and highly relevant to real-world forecasting systems, where retraining large backbones is often infeasible. The proposed δ-Adapter is simple yet elegant, introducing bounded input and output modules that deliver measurable gains without altering the base model. Theoretical sections are rigorous, offering provable guarantees of local descent, drift stability, and compositional safety, which is an uncommon level of formalism for this kind of lightweight method. The paper is also clear and well-structured, with intuitive explanations of each component and thoughtful use of figures to convey mechanisms. Empirically, the approach demonstrates consistent and transferable improvements across architectures and datasets with negligible overhead, highlighting its significance and applicability to a broad range of forecasters. Overall, the work stands out for its conceptual clarity, strong theoretical underpinnings, and practical impact on efficient and reliable time-series forecasting.

**Weaknesses:**

While the paper presents a strong theoretical foundation and an appealing practical idea, several areas could be improved to strengthen its impact and clarity.

First, the experimental evaluation, though broad in dataset coverage, lacks comparative depth. The study primarily benchmarks δ-Adapter against frozen and fine-tuned baselines, but it omits comparisons to parameter-efficient adaptation methods such as LoRA, adapters, or residual fine-tuning strategies that have been explored in related forecasting and NLP contexts. Including such baselines would better position δ-Adapter within the broader adaptation literature and clarify its relative advantages.

Second, while the theoretical analysis is thorough, the empirical link to these guarantees remains weak. The paper could provide visual or quantitative evidence of stability, such as loss landscapes or δ–performance trade-offs, to demonstrate how the theory manifests in practice.

Third, the related work section should be expanded to discuss recent advances in adaptation and test-time adaptation (TTA) for time-series forecasting, which directly address similar motivations to δ-Adapter. In particular, works such as Kim et al. (AAAI 2025) [1], Medeiros et al. (arXiv 2025) [2], Grover & Etemad (ICML 2025 Workshop) [3], and Lee et al. (ICML 2025) [4] explore gradient-based, parameter-efficient, and online adaptation mechanisms to handle non-stationarity and distribution shifts. Adding a dedicated subsection contrasting δ-Adapter with these methods would clearly position it as a post-hoc alternative and highlight its novelty within the adaptation landscape.

Overall, the paper would benefit from stronger empirical grounding, richer comparative analysis, and a more comprehensive connection to existing adaptation research to fully demonstrate the breadth and impact of its contribution.

References:

[1] Kim, H., Kim, S., Mok, J., & Yoon, S. (2025, April). Battling the non-stationarity in time series forecasting via test-time adaptation. In Proceedings of the AAAI Conference on Artificial Intelligence (Vol. 39, No. 17, pp. 17868-17876).

[2] Medeiros, H. R., Sharifi-Noghabi, H., Oliveira, G. L., & Irandoust, S. (2025). Accurate Parameter-Efficient Test-Time Adaptation for Time Series Forecasting. arXiv preprint arXiv:2506.23424.

[3] Grover, Shivam, and Ali Etemad. "Shift-Aware Test Time Adaptation and Benchmarking for Time-Series Forecasting." Second Workshop on Test-Time Adaptation: Putting Updates to the Test! at ICML 2025. 2025.

[4] Lee, Thomas L., et al. "Lightweight Online Adaption for Time Series Foundation Model Forecasts." Forty-second International Conference on Machine Learning.

**Questions:**

1. **Comparison to Test-Time Adaptation (TTA) Methods**.
   How does δ-Adapter conceptually and empirically differ from recent TTA approaches that also adapt frozen forecasters at inference time (e.g., Kim et al., AAAI 2025; Medeiros et al., 2025; Grover & Etemad, ICML 2025)? Since these methods share similar goals such as robustness to non-stationarity and post-hoc improvement, please clarify what specific challenges δ-Adapter addresses that are not tackled by gradient-based or parameter-efficient TTA methods.

2. **Choice and Sensitivity of δ**.
   The theoretical analysis emphasizes δ as a small trust-region parameter controlling stability. How sensitive are results to δ in practice, and how should it be chosen for new datasets or models? Could adaptive δ-scheduling be beneficial?

3. **Theoretical–Empirical Link**.
   The paper provides strong theoretical guarantees for stability and local descent, but the experiments do not directly validate these properties. Could the authors include empirical evidence such as δ vs. loss curves or visualization of stability bounds to support the theoretical claims?

4. **Comparison to Lightweight Adaptation Baselines**.
   Have the authors considered comparing δ-Adapter with parameter-efficient fine-tuning strategies such as LoRA or standard adapter layers, which also introduce small modules on frozen backbones? This would help quantify how δ-Adapter performs relative to existing low-overhead adaptation techniques.

5. **Practical Deployment Considerations**.
   Since δ-Adapter is advertised as a post-hoc enhancement for deployed models, what is the typical training cost, latency overhead, and implementation effort in a real-world deployment scenario? Including such details would reinforce the paper’s practical relevance.

These clarifications would significantly help assess the robustness, novelty, and real-world applicability of δ-Adapter and could strengthen the case for acceptance.

---

> ### Author Response · Authors · 2025-11-20
> **Response to Reviewer Fzqc [Part 1]**
>
> We would like to sincerely thank Reviewer Fzqc for recognition of our strong theoretical foundation and an appealing practical idea, as well as for providing the insightful review and constructive suggestions. **We have incorporated the weaknesses into the questions block to provide an overall response.** Below are the responses point by point:
>
> - **Q1: Comparison to Test-Time Adaptation (TTA) Methods.**
>
> A1-Part 1: **Differ from recent TTA approaches:** The existing TTA approaches aim to mitigate test-time concept drift via selective layer retraining (SOLID), online linear adapter updates (TAFAS) and its follow-ups PETSA (auxiliary loss) and DynaTTA (dynamic gating), and parallel forecaster combine (ELF), layer-wise adjustment and memory (FSNet), and dynamic model selection (OneNet). These methods are either based on linear adapters, parallel fusion, or overall fine-tuning; and, they do not consider the impact of label leakage ([Liang et al. 2024](https://arxiv.org/abs/2412.00108), [YA Lau et al. 2025](https://openreview.net/pdf?id=I0n3EyogMi)). *$\delta$-Adapter can perform non-linear adaptation on both input and output, with good theoretical guarantees. And it only relies on the most recent sample for fast updates. In addition, it can be used as a feature selector or a corrector.*
>
> A1-Part 2: **Differ from fine-tuning approaches in NLP:** LoRA-style adapters in NLP tend to lead to high performance variance, *since the output range is not fixed (classification task, with probabilities in 0-1), the model tends to exhibit high variance (parameter changes directly affects the outcomes)*, as shown in Figure 8(c).  However, $\delta$-Adapter differs from existing adapters in NLP in the following aspects:
> - - $\delta$-Adapter is a post-hoc I/O editor of black-box forecasters. But, NLP adapters (e.g., parameter-efficient fine-tuning in Transformers) are typically inserted inside each layer and trained jointly with full access to the backbone internals.
> - - NLP adapters do not modify the I/O data themselves, but rather add additional infos/params to them or calculates auxiliary losses or penalties to achieve domain alignment. In contrast, $\delta$-Adapter places more emphasis on the gains brought to the model by editing or correcting the I/O data itself.
>
> Alternatively, applying $\delta$-Adapter to NLP is also a promising attempt. We have added comparisons in Appendix A of the new version.
>
> A1-Part 3: **Compared the post-processing methods.** For fair comparison, we removed label leakage [Liang et al. 2024](https://arxiv.org/abs/2412.00108) and [YA Lau et al. 2025](https://openreview.net/pdf?id=I0n3EyogMi). Our experiments show that our method achieves the lowest error on every dataset across all backbones (**Tables 2 & 10 in new version**).
> ||DistPred|+SOLID|+TAFAS|+LoRA|+Ours|iTransformer|+SOLID|+TAFAS|+LoRA|+Ours|
> |:-:|:-:|:-:|:-:|:-:|:-:|:-:|:-:|:-:|:-:|:-:|
> |ELC|0.182|0.182|0.182|0.18|**0.175**|0.19|0.19|0.19|0.186|**0.18**|
> |ETTh1|0.461|0.46|0.476|0.454|**0.451**|0.454|0.458|0.477|0.448|**0.449**|
> |ETTh2|0.39|0.391|0.402|0.385|**0.379**|0.388|0.393|0.448|0.384|**0.377**|
> |ETTm1|0.412|0.406|0.411|0.407|**0.396**|0.417|0.414|0.42|0.414|**0.403**|
> |ETTm2|0.285|0.285|0.288|0.281|**0.274**|0.3|0.298|0.304|0.293|**0.29**|
> |Exange|0.35|0.347|0.363|0.336|**0.297**|0.383|0.376|0.392|0.376|**0.316**|
> |Traffic|0.453|0.453|0.455|0.449|**0.44**|0.475|0.475|0.476|0.468|**0.461**|
> |Weather|0.256|0.255|0.256|0.251|**0.242**|0.259|0.257|0.259|0.255|**0.244**|
>
> A1-Part 4: **What challenges did $\delta$-Adapter address:** $\delta$-Adapter is insensitive to the learning rate, easy to converge, and has low variance since it directly limits the editing range of I/O, as shown in Fig. 8(b).
> In summary, **the challenges addressed by $\delta$-Adapter include:**
> - - Conditions drift, which refers to gradual changes in the data-generating process (e.g., seasonal regime shifts, covariate shifts in demand patterns) that occur after the model has been deployed, making full retraining costly;
> - - High performance variance. Existing post-processing techniques are prone to have high performance variance due to unstable training;
> - - Inefficient training/inference. Using complex modules or frequent updates to absorb low-complexity residuals makes models suffer from inefficient training/inference.
>
>  **The benefits of the post-processing module are**:
> - - A plug-and-play, model-agnostic lightweight module that can be used for untouchable black-box models and brings significant performance improvements;
> - - Freezing backbone avoids catastrophic forgetting and robust to high performance variance, preserves production hardening and governance;
> - - Fast training/test speeds and high data efficiency are achieved, even with short labeled histories or tight retraining windows;
> - - Metric-aware, it enables important feature selection and application-aligned calibration without altering the backbone.

---

> ### Author Response · Authors · 2025-11-20
> **Response to Reviewer Fzqc [Part 2]**
>
> ---
> - **Q2: Choice and Sensitivity of $\delta$.**
>
> A2-Part 1: **Sensitivity of $\delta$.** Ablation studies of $\delta$ are shown in below. It can be found that when $\delta\approx0.1$, a decent performance can be achieved.
> |$\delta$=|0.01|0.05|0.1|0.2|0.3|
> |:-:|:-:|:-:|:-:|:-:|:-:|
> |ELC|0.163|0.161|0.160|0.160|0.161|
> |Weather | 0.171  | 0.170  | 0.172  |0.177|0.182|
> |Traffic|0.443|0.439|0.435|0.432|0.433|
>
> A2-Part 2: **Choice of $\delta$.** $\delta$ is related to the properties of the dataset (e.g., noise level, degree of concept drift). In our work, we divided the datasets into two categories: one with severe concept drift ($\delta=0.1$, e.g., Traffic, Weather, etc.) and the other with non-severe concept drift ($\delta=0.01$, e.g., ETT, etc.). For datasets with severe concept drift, setting $\delta=0.1$ is sufficient. As shown in the table below (Page 27, Table 12), a better value of $\delta=0.1$ might yield better results. In our paper, we only reported the two settings ($\delta$=0.1 or 0.01).
> |+0.1X||x0.1X||x0.2X||+0.1Y||x0.1Y||x0.2Y||+0.1(X&Y)||x0.1(X&Y)||x0.2(X&Y)||
> |:-:|:-:|:-:|:-:|:-:|:-:|:-:|:-:|:-:|:-:|:-:|:-:|:-:|:-:|:-:|:-:|:-:|:-:|
> |Val|Test|Val|Test|Val|Test|Val|Test|Val|Test|Val|Test|Val|Test|Val|Test|Val|Test|
> |0.418|0.166|0.427|0.168|0.425|0.169|0.421|0.168|0.415|0.165|0.413|0.167|0.413|0.160|0.416|0.162|0.411|0.162|
>
> A2-Part 3: **Could adaptive $\delta$-scheduling be beneficial?** Adaptive $\delta$-scheduling is definitely beneficial and a very promising direction. However, a systematic study of such policies is orthogonal to our main contributions. We believe it is an interesting extension that could further exploit the $\delta$-Adapter framework.
>
> - **Q3: Theoretical–Empirical Link, e.g., $\delta$ vs. loss curves or visualization of stability bounds.**
>
> A3: Below, we present the performance changes of additive and multiplicative adapters on different datasets over epochs (the blank is due to early stopping). In **Figure 9 (Page 24)** of the new version, we visualize the changes in validation and test losses of $\delta$-Adapter across different datasets.
> ||Original|1|2|3|4|5|6|7|8|9|10|
> |:-:|:-:|:-:|:-:|:-:|:-:|:-:|:-:|:-:|:-:|:-:|:-:|
> |ELC (Add)|0.167|0.161|0.158|0.159|0.159|0.159|0.159|0.159|0.159|0.159|0.159|
> |ELC (Mul)|0.167|0.162|0.161|0.160|0.159|0.159|0.159|0.159|0.159|||
> |Traffic (Add)|0.463|0.460|0.455|0.453|0.452|0.451|0.451|0.451|0.451|0.451|0.451|
> |Traffic (Mul)|0.463|0.459|0.454|0.451|0.449|0.448|0.448|0.448|0.448|0.448||
> |Weather (Add)|0.178|0.162|0.161|0.161|0.161|0.161|0.161|0.161|0.161|0.161||
> |Weather (Mul)|0.178|0.168|0.166|0.166|0.165|0.165|0.165|0.165|0.165|0.165|0.165|
>
> These experiments show that the loss curve of the $\delta$-Adapter gradually decreases with epochs and has stable and consistent boundaries. Meanwhile, as shown in the table in A2-Part 2, our composite adapter (X&Y) can achieve better performance (Stability Analysis of Section 3), which also proves the robustness of the $\delta$-Adapter and the correctness of its theoretical foundation.
>
> - **Q4: Comparison to Lightweight Adaptation Baselines, e.g., LoRA.**
>
> A4:  We compared recent lightweight adaptation baselines, which aim to mitigate test-time concept drift via selective layer retraining (SOLID), online adapter updates (TAFAS),  auxiliary loss (PETSA), layer-wise adjustment and memory (FSNet), and dynamic model selection (OneNet). However, according to works by [Liang et al. 2024](https://arxiv.org/abs/2412.00108) and [YA Lau et al. 2025](https://openreview.net/pdf?id=I0n3EyogMi), the above methods have used future labels to some extent, causing label leakage in long-term forecasting, where future ground truth is adopted in advance for adaptation. For fair comparison, we removed label leakage (it may cause performance degradation of some adapters):
>
> ||DistPred|+SOLID|+TAFAS|+LoRA|+Ours|iTransformer|+SOLID|+TAFAS|+LoRA|+Ours|
> |:-:|:-:|:-:|:-:|:-:|:-:|:-:|:-:|:-:|:-:|:-:|
> |ELC|0.182|0.182|0.182|0.18|**0.175**|0.19|0.19|0.19|0.186|**0.18**|
> |ETTh1|0.461|0.46|0.476|0.454|**0.451**|0.454|0.458|0.477|0.448|**0.449**|
> |ETTh2|0.39|0.391|0.402|0.385|**0.379**|0.388|0.393|0.448|0.384|**0.377**|
> |ETTm1|0.412|0.406|0.411|0.407|**0.396**|0.417|0.414|0.42|0.414|**0.403**|
> |ETTm2|0.285|0.285|0.288|0.281|**0.274**|0.3|0.298|0.304|0.293|**0.29**|
> |Exange|0.35|0.347|0.363|0.336|**0.297**|0.383|0.376|0.392|0.376|**0.316**|
> |Traffic|0.453|0.453|0.455|0.449|**0.44**|0.475|0.475|0.476|0.468|**0.461**|
> |Weather|0.256|0.255|0.256|0.251|**0.242**|0.259|0.257|0.259|0.255|**0.244**|
>
> Our experiments show that our method achieves the lowest error on every dataset across all backbones (**Tables 2 & 10 in new version**).

---

> ### Author Response · Authors · 2025-11-20
> **Response to Reviewer Fzqc [Part 3]**
>
> Q5: Practical Deployment Considerations, e.g., training cost, latency overhead, and implementation effort.
>
> A5-Part 1: The training/test time, parameters and memory are:
>
> ||iTransformer||+Ours ~3M||+SOLID ~0.5M||+TAFAS ~6M||OneNet ~3M||FSNet ~2M||
> |:-:|:-:|:-:|:-:|:-:|:-:|:-:|:-:|:-:|:-:|:-:|:-:|:-:|
> |Length|Time (S)|Memory (MB)|Time|Memory|Time|Memory|Time|Memory|Time|Memory|Time|Memory|
> |96|38|460|**167**|1696|554|476|852|2026|444|722|231|704|
> |192|43|462|**175**|2402|497|483|865|2270|406|738|192|700|
> |336|47|463|**201**|3312|458|486|1104|2648|363|740|194|700|
> |720|60|464|**217**|5792|293|502|2142|5672|275|742|163|700|
>
> The above table shows that $\delta$-Adapter is the most time-efficient, it is consistently faster than other methods across all horizons: Taking the offline method as a benchmark, our proposed method **(Ada-X+Y) achieves the fastest speed**. Therefore, $\delta$-Adapter has more deployment advantages due to its simplicity and powerful nature.
>
> We trust that these clarifications regarding the distinct advantages of our post-hoc approach adequately address the reviewers' concerns. Many thanks!

---

> > ### Author Response · Authors · 2025-12-02
> > **Response to Reviewer Fzqc [Part 4]: Supplementary to Q5**
> >
> > **Q5: Practical Deployment Considerations, e.g., training cost, latency overhead, and implementation effort.**
> >
> > A5-Part 2: **Training cost and latency overhead.**
> >
> > The $\delta$-Adapter is designed to be extremely lightweight. It is implemented as a 2-layer MLP (specifically, for **Sundial (128M)** and **TabPFN (48M)**, the adapter adds only approx. 1.5M-3M parameters depending on the horizon). Compared to the backbone model, the adapter introduces **less than 2%-6%** additional parameters, validating the lightweight claim.
> >
> > A4-Part 2: **Training cost and latency overhead**
> >
> > We compared the wall-clock time of our method against state-of-the-art test-time adaptation (TTA) methods. As shown in the table below (and Table 6 in the revision), while online adaptation naturally incurs overhead compared to a static offline model (which performs no updates), our proposed method **(Ada-X+Y) is consistently the fastest among all adaptive methods**. When used offline, the additional computational overhead of $\delta$-Adapter is negligible.
> >
> > *Table 6: Time (S) and memory (MB) of adapters (backbone is TabPFN) and online methods.*
> > |Offline ~48M||+Ours ~3M|||+SOLID ~0.5M|||+TAFAS ~6M|||+OneNet ~3M|||+FSNet ~2M|||
> > |:-:|:-:|:-:|:-:|:-:|:-:|:-:|:-:|:-:|:-:|:-:|:-:|:-:|:-:|:-:|:-:|:-:|
> > |Time|Memory|Train|Test|Memory|Train|Test|Memory|Train|Test|Memory|Train|Time|Memory|Train|Time|Memory|
> > |281|1840|**392**|**395**|1983|511|667|2401|603|861|3468|693|471|1512|621|485|1504|
> > |307|1848|**386**|**379**|2132|481|624|2423|589|895|3790|681|445|1537|618|472|1531|
> > |326|1852|**385**|**415**|2622|484|593|2446|583|1152|4186|631|452|1559|599|466|1517|
> > |351|1856|**369**|**431**|3102|505|398|2501|916|1803|6809|530|465|1567|554|458|1526|
> >
> > It is worth noting that for some methods in above table, due to reasons such as backprop gradients, the time is related to the number of sample updates, i.e., the longer the prediction length, the fewer the samples, and the shorter the time.
> >
> > A4-Part 3: Implementation Details
> >
> > $\delta$-Adapter can be divided into input editing (Ada-X), output correction (Ada-Y), and their composite forms (Ada-X+Y).
> > The forward pass is:
> > $$\hat Y=F(X + \delta A_{\theta}^{in}(X)) \qquad (1)$$
> > $\tilde Y = \hat Y + \delta A_{\theta}^{out}(\hat Y)\qquad(2)$
> >
> > - - For Ada-X, gradients flow from the loss through the backbone $F$, then through Ada-X (Eq. 2).
> > - - For Ada-Y, gradients flow from the loss through Ada-Y (Eq. 2), then through the backbone $F$.
> > -- For Ada-X+Y, Ada-X and Ada-Y are **trained jointly** in an end-to-end manner, not sequentially. We minimize **a single combined loss** $\mathcal{L}$ over the union of parameters $A_{\theta}^{in}$ (Ada-X) and $A_{\theta}^{out}$ (Ada-Y).
> > During the backward pass, gradients flow from the loss through Ada-Y (Eq. 2), then through the backbone $F$, and finally to Ada-X (Eq. 1). This ensures that Ada-X learns input perturbations that specifically help the backbone produce features that Ada-Y can best correct.
> >
> > Experimental Setup: In our codebase, we instantiate two separate Adam optimizers (learning rate=1E-4) for modular flexibility. However, they are stepped simultaneously after a single backward pass, making the process equivalent to optimizing a joint objective. As derived in **Proposition 3.2**, this joint update rule maintains the $O(\delta)$ drift bounds and descent guarantees, ensuring the two adapters do not destabilize each other (**See Appendix C.4 of the new version**).
> >
> > In real-world deployment, $\delta$-Adapter introduces negligible latency. Since the backbone $F$ remains frozen, the inference process involves only a standard forward pass of $F$ plus the lightweight forward pass of the adapter $A_\theta$ (a shallow MLP). Furthermore, for the output-side adapter, implementation is strictly black-box compatible, allowing improvements to deployed APIs (e.g., TabPFN, Sundial and TimesFM, etc.) without access to model internals (**see Appendix C.12 & Tabel 13**).
> >
> > We hope these responses adequately address your concerns. Thanks.

---

### Official Review · Reviewer_gkGB · 2025-10-31

**Soundness:** 3
**Presentation:** 3
**Contribution:** 3
**Rating:** 8
**Confidence:** 4

**Summary:**

This paper presents an adapter-based finetuning method for pretrained forecasting models. The paper proposes input and output-based adapters in two forms: additive and multiplicative. The authors provide theoretical results for the stability and improvement under certain conditions with respect to the base model.  Finally, the authors expose how to utilize their adapters as calibration methods for models that only offer fixed-point predictions.

**Strengths:**

- Overall, I think this is a good paper. The presentation is clear and the theory is solid as far as I can tell.
- The presented experiments consistently show the improvement of input and output adapters.
- Benchmarks look reasonable within popular datasets from the time series literature.

**Weaknesses:**

- Some of the models are relatively old and would not be considered SOTA at the current time (e.g. Autoformer). I would suggest including models like PatchTST, Non-stationary Transformer, TimesNet or TimeMixer.
- Additive vs multiplicative:
- Reports metrics in Table 2 are  averaged across lenghts. I understand the difficulty of reporting metrics across many horizons, but in my opinion, this makes it harder to interpret the effect over prediction lengths, over which errors may differ significantly.

**Typos and minor comments**
- L132: "Obviously". This can be dropped :)
- Table 2 Exange (also in the appendix table). caption: "veraged"
- L64: Despite adapter is simplicity

**Questions:**

- L461-463: Could the authors elaborate more on why this type of input/output adapter is desirable over a LoRA style adapter?
- How does additive vs multiplicative updates compare empirically? It seems the experiments mostly focus on additive adapters.
- Why not report results for Ada X+y adapters together (Tables 1 and 2)?

---

> ### Author Response · Authors · 2025-11-20
> **Response to Reviewer gkGB**
>
> We thank the Reviewer gkGB for the positive evaluation and valuable comments. Below are the responses point by point:
>
> - **W1: Some of the models are relatively old.**
>
> A1：We have introduced the mentioned works in the new version and selected the models with SOTA performance among them (PatchTST and TimeMixer) as the backbone to test $\delta$-Adapter (**Page 9 and Table 5 of the new version**). The results are as follows:
> ||PatchTST||+ Ada-X+Y||+ Ada-X$\times$Y||TimeMixer||+ Ada-X+Y||+ Ada-X$\times$Y||
> |:-:|:-:|:-:|:-:|:-:|:-:|:-:|:-:|:-:|:-:|:-:|:-:|:-:|
> ||MSE|MAE|MSE|MAE|MSE|MAE|MSE|MAE|MSE|MAE|MSE|MAE|
> |ELC|0.167|0.252|**0.159**|0.245|**0.159**|0.246|0.145|0.243|**0.143**|0.241|**0.143**|0.241|
> |Weather|0.178|0.219|**0.161**|0.22|*0.165*|0.224|0.168|0.216|*0.166*|0.214|**0.164**|0.214|
> |Traffic|0.463|0.297|*0.451*|0.292|**0.448**|0.29|0.475|0.317|**0.465**|0.307|*0.467*|0.31|
>
> Obviously, after adding the $\delta$-Adapter to PatchTST and TimeMixer, their performance has been significantly improved.
>
> - **W2: The average indicators in Table 2 make it difficult to explain the impact of predicted length.**
>
> A2: Table 2 is the average of Table 9 (new version) over multiple prediction lengths. We have added a link to Table 9 in Table 2 to make it easier to explain the impact on prediction lengths.
>
> - **W3: Typos and minor comments.**
>
> A3: Thanks. We have corrected the mentioned contents.
>
> > **Q1: L461-463: Could the authors elaborate more on why this type of input/output adapter is desirable over a LoRA style adapter?**
>
> A4: LoRA-style adapters tend to lead to high performance variance, as shown in Figure 8(c) of our paper. This phenomenon occurs because LoRA-style adapters are generally used for classification (NLP), where the output range of the model is fixed (probabilities in 0-1), so the model is less likely to collapse. However, when LoRA is applied to regression, *since the output range is not fixed, the model tends to exhibit high variance (parameter changes directly affects the outcomes)*. In such cases, it is necessary to carefully select hyperparameters such as the learning rate to ensure that LoRA is fine-tuned within the original output space, which is not easy.
>
> On the contrary, because $\delta$-Adapter directly limits the editing range of I/O, it is insensitive to the learning rate, and is easy to converge and has low variance, as shown in Fig. 8(b).
> In summary, the benefits of the post-processing module include:
> - - It is a plug-and-play, model-agnostic lightweight module that can bring significant performance gains;
> - - Freezing backbone avoids catastrophic forgetting and robust to overfitting, preserves production hardening and governance;
> - - Training a tiny adapter is fast and data-efficient, even with short labeled histories or tight retraining windows;
> - - It can be metric-aware, enabling application-aligned calibration without touching backbone.
>
> > **Q2: How does additive vs multiplicative updates compare empirically? It seems the experiments mostly focus on additive adapters.**
>
> A5：In the table in **A1** above, the additive and multiplicative adapters reduce the MSE of *PatchTST* by **5.6%** and **5.1%** respectively across various datasets. However, for *TimeMixer*, the MSE reductions from the additive and multiplicative adapters are **1.6%** and **1.8%** respectively. This indicates that both have their respective advantages, and the increase of the additive adapter is relatively more significant.
> |+0.1X||x0.1X||x0.2X||+0.1Y||x0.1Y||x0.2Y||+0.1(X&Y)||x0.1(X&Y)||x0.2(X&Y)||
> |:-:|:-:|:-:|:-:|:-:|:-:|:-:|:-:|:-:|:-:|:-:|:-:|:-:|:-:|:-:|:-:|:-:|:-:|
> |Val|Test|Val|Test|Val|Test|Val|Test|Val|Test|Val|Test|Val|Test|Val|Test|Val|Test|
> |0.418|0.166|0.427|0.168|0.425|0.169|0.421|0.168|0.415|0.165|0.413|0.167|0.413|0.160|0.416|0.162|0.411|0.162|
>
> In addition, we conducted experiments on additive and multiplicative adapters with different $\delta$ on input/output. As shown in the results above, *it can be seen that both additive and multiplicative adapters can achieve significant gains, while additive adapters are relatively more robust than multiplicative ones.*
>
> > **Q3: Why not report results for Ada X+Y adapters together (Tables 1 and 2)?**
>
> A6: In Tables 1 and 2 we initially focused on Ada-X and Ada-Y separately in order to isolate the effect of editing the input vs. the output of a frozen forecaster and to keep the main tables compact. The combined adapter (Ada-X+Y) is evaluated in our ablation section (Figure 2 and Table 5), where it consistently achieves the lowest MSE among all variants under both batch and online training.
>
> We hope these responses adequately address your concerns. Thank you for your valuable suggestions.

---

### Official Review · Reviewer_5yqn · 2025-11-01

**Soundness:** 3
**Presentation:** 3
**Contribution:** 3
**Rating:** 6
**Confidence:** 3

**Summary:**

This paper introduces sigma-Adapter, a lightweight post-processing framework for improving time series forecasting without retraining the backbone model. The approach works by learning small bounded corrections at two interfaces: (1) input nudging (soft edits to covariates) and (2) output residual correction. The authors provide theoretical guarantees for local descent and stability and demonstrate feature selection capabilities through learnable masks, and introduce two distributional correctors (Quantile and Conformal calibrators). The experiments across multiple datasets and backbone models show consistent improvements in both accuracy and calibration.

**Strengths:**

The main strengths of the paper include,
1. The post-processing approach is a well-motivated solution for real-world deployments where retraining large models is costly. The ability to improve frozen forecasters is important and useful problem to tackle.
2. The paper provides rigorous theoretical analysis for several algorithms: Local descent guarantees (Theorems 2 & 3), Compositional stability for combined adapters (Proposition 3.2) etc.
3. The authors compared against diverse backbones (DistPred, iTransformer, FourierGNN, FreTS, Autoformer) across several datasets (ETT, Traffic, Weather, Electricity, Exchange). Both pre-trained and SOTA models show consistent improvements in both accuracy and calibration.
4. The authors conducted clear ablation studies to 1. Compare the performance of sigma-adapter variants. 2. Compare the effectiveness of input adapters and output calibrators across several datasets.

**Weaknesses:**

The main weakness of the paper include,

1. The proposed Input/output adapters approach is conceptually similar to existing adapter methods in NLP. The theoretical results (gradient descent, Lipschitz stability) are relatively standard. The main contribution does not appear to be novel as the claims in the paper and looks like an application of the NLP concept to time series.
2. Some important ablation studies such as the impact of adapter architecture (MLP depth, width etc), effect of different sigma values across datasets are missing.
3. The comparison with other post-processing method (e.g., calibration techniques) and adapter approaches are not presented; The comparison with lightweight fine-tuning methods (e.g., LoRA for time series) across multiple pre-trained models is missing; The LoRA comparison in Figure 8(c) is limited to one model (Sundial) and shows high variance but needs more investigation.
4. Discussion on fine-tuning/adaptation time across different adapter approaches and comparison against wider range of pretrained models might be helpful to strengthen this paper.

**Questions:**

1. Can you compare against other adapter approaches available in time series literature and also compare with adapters used in NLP literature?
2. Can you have thorough discussion on the comparison with LoRA method and investigation on high variance?
3. Can you compare against few other pre-trained models like TabPFN-TS to validate the claims made are generalizable across pre-trained models?

---

> ### Author Response · Authors · 2025-11-20
> **Response to Reviewer 5yqn [Part 1]**
>
> We would like to sincerely thank Reviewer 5yqn for providing the insightful review and constructive suggestions. Below are the responses point by point:
>
> - **W1: Novelty of the proposed method and theoretical results.**
>
> A1-Part 1: **Novelty of the proposed method.** The proposed $\delta$-Adapter is novel compared to existing methods. The $\delta$-Adapter differs from existing adapters in NLP in the following aspects (**in Pages 1 & 17 of the new version**):
> - - $\delta$-Adapter is a post-hoc I/O editor of black-box forecasters. But, NLP adapters (e.g., parameter-efficient fine-tuning in Transformers) are typically inserted inside each layer and trained jointly with full access to the backbone internals.
> - - NLP adapters (prompt, prefix, P-tuning, Lora) do not modify the I/O data themselves, but rather add additional infos/params to them or calculates auxiliary losses or penalties to achieve domain alignment. In contrast, $\delta$-Adapter places more emphasis on the gains brought to the model by editing or correcting the I/O data itself.
>
> **The benefits of the post-processing module are**:
> - - A plug-and-play, model-agnostic lightweight module that can be used for untouchable black-box models and brings significant performance improvements;
> - - Freezing backbone avoids catastrophic forgetting and robust to high variance, preserves production hardening and governance;
> - - Fast training/test speeds and high data efficiency are achieved, even with short labeled histories or tight retraining windows;
> - - Metric-aware, it enables important feature selection and application-aligned calibration without altering the backbone.
>
> To the best of our knowledge, prior adapter works in NLP do not cover this combination of post-hoc I/O manipulation, feature selection, and calibrated predictive intervals, all within a single unified framework. Alternatively, applying $\delta$-Adapter to NLP is also a promising attempt. We have added comparisons in Appendix A of the new version.
>
> A1-Part 2: **Novelty of the theoretical results.** Our theoretical results (Proposition 3.1, Theorems 2–3, and Proposition 3.2) provide: 1) explicit Lipschitz drift bounds for input edits and their multiplicative variant, 2) per-sample local descent guarantees for both output and input adapters, and 3) a compositional stability result when combining input and output adapters.
> Below, we present the performance changes of $\delta$-adapters on different datasets over epochs (the blank is due to early stopping):
> ||Original|1|2|3|4|5|6|7|8|9|10|
> |:-:|:-:|:-:|:-:|:-:|:-:|:-:|:-:|:-:|:-:|:-:|:-:|
> |ELC (Add)|0.167|0.161|0.158|0.159|0.159|0.159|0.159|0.159|0.159|0.159|0.159|
> |ELC (Mul)|0.167|0.162|0.161|0.160|0.159|0.159|0.159|0.159|0.159|||
> |Traffic (Add)|0.463|0.460|0.455|0.453|0.452|0.451|0.451|0.451|0.451|0.451|0.451|
> |Traffic (Mul)|0.463|0.459|0.454|0.451|0.449|0.448|0.448|0.448|0.448|0.448||
> |Weather (Add)|0.178|0.162|0.161|0.161|0.161|0.161|0.161|0.161|0.161|0.161||
> |Weather (Mul)|0.178|0.168|0.166|0.166|0.165|0.165|0.165|0.165|0.165|0.165|0.165|
>
> In Figure 9 of  new version of the paper, we also visualize the changes in validation and test losses of $\delta$-Adapter across different datasets.
> These experiments show that the loss curve of the $\delta$-Adapter gradually decreases with epochs and has stable and consistent boundaries. Meanwhile, as shown in Fig. 7, our composite adapter (X&Y) can achieve better performance (Stability Analysis of Section 3), which also proves the robustness of the $\delta$-Adapter and the correctness of its theoretical foundation.
>
> - **W2:  Ablation studies of $\delta$-Adapter's depth, width and $\delta$.**
>
> A2: Ablation studies of $\delta$-Adapter's depth and width are shown below. It can be found that the depth has little impact on performance, while the greater the width, the slight improvement in performance (**in Page 26 of the new version**).
> |Depth|2||||3||||4||||
> |:-:|:-:|:-:|:-:|:-:|:-:|:-:|:-:|:-:|:-:|:-:|:-:|:-:|
> |Width|64|128|256|512|64|128|256|512|64|128|256|512|
> |ELC|0.159|0.157|0.155|0.1533|0.158|0.157|0.154|0.152|0.159|0.157|0.154|0.152|
> |Weather|0.162|0.16|0.159|0.158|0.162|0.161|0.159|0.158|0.162|0.161|0.16|0.158|
> |Traffic|0.439|0.437|0.433|0.43|0.44|0.436|0.433|0.43|0.44|0.437|0.434|0.43|
>
> Ablation studies of $\delta$ are shown in below. It can be found that when $\delta$ is approximately 0.1, a decent performance can be achieved.
> |$\delta$=|0.01|0.05|0.1|0.2|0.3|
> |:-:|:-:|:-:|:-:|:-:|:-:|
> |ELC|0.163|0.161|0.160|0.160|0.161|
> |Weather|0.171|0.170|0.172|0.177|0.182|
> |Traffic|0.443|0.439|0.435|0.432|0.433|

---

> ### Author Response · Authors · 2025-11-20
> **Response to Reviewer 5yqn [Part 2]**
>
> - **W3: Comparison with other post-processing methods, e.g., calibration techniques and LoRA.**
>
> A3: We compared recent post-processing methods, which aim to mitigate test-time concept drift via selective layer retraining (SOLID), online adapter updates (TAFAS),  auxiliary loss (PETSA), layer-wise adjustment and memory (FSNet), and dynamic model selection (OneNet). However, according to works by [Liang et al. 2024](https://arxiv.org/abs/2412.00108) and [YA Lau et al. 2025](https://openreview.net/pdf?id=I0n3EyogMi), the above methods have used future labels to some extent, causing label leakage in long-term forecasting, where future ground truth is adopted in advance for adaptation. For fair comparison, we removed label leakage (it may cause performance degradation of some adapters):
>
> ||DistPred|+SOLID|+TAFAS|+LoRA|+Ours|iTransformer|+SOLID|+TAFAS|+LoRA|+Ours|
> |:-:|:-:|:-:|:-:|:-:|:-:|:-:|:-:|:-:|:-:|:-:|
> |ELC|0.182|0.182|0.182|0.18|**0.175**|0.19|0.19|0.19|0.186|**0.18**|
> |ETTh1|0.461|0.46|0.476|0.454|**0.451**|0.454|0.458|0.477|0.448|**0.449**|
> |ETTh2|0.39|0.391|0.402|0.385|**0.379**|0.388|0.393|0.448|0.384|**0.377**|
> |ETTm1|0.412|0.406|0.411|0.407|**0.396**|0.417|0.414|0.42|0.414|**0.403**|
> |ETTm2|0.285|0.285|0.288|0.281|**0.274**|0.3|0.298|0.304|0.293|**0.29**|
> |Exange|0.35|0.347|0.363|0.336|**0.297**|0.383|0.376|0.392|0.376|**0.316**|
> |Traffic|0.453|0.453|0.455|0.449|**0.44**|0.475|0.475|0.476|0.468|**0.461**|
> |Weather|0.256|0.255|0.256|0.251|**0.242**|0.259|0.257|0.259|0.255|**0.244**|
>
> Our experiments show that our method achieves the lowest error on every dataset across all backbones (**Tables 2 & 10 in new version**).
>
> - **W4: Discussion different adapters and comparison against wider range of pretrained models.**
>
> A4: The discussion on $\delta$-Adapter and NLP adapters can be found in **A1**, and the comparison among $\delta$-Adapter and other adapter-based methods on different pre-trained models can be found in **A3** (pre-trained models are DistPred, iTransformer) and **A7** (pre-trained model is TabPFN-TS). The revisions of the paper include the Introduction, Experiments, and Recent work sections. These are also included in the Introduction, Experiments, and Related work sections of the new version PDF.
>
> ---
>
> > **Q1: Can you compare against other adapter approaches available in time series literature and also compare with adapters used in NLP literature?**
>
> A5: We compared recent adapter approaches in time series, including SOLID, TAFAS (follow-up methods like PETSA and DynaTTA perform similarly to TAFAS), and LoRA-style methods in NLP, in the above **A3**. It shows that our method achieves the lowest error on every dataset across all backbones (**Tables 2 & 10 in new version**). Furthermore, we discussed the differences between $\delta$-Adapter and the adapters in NLP in **A1**, and emphasized this in the Introduction (Page 1), Experiments (Page 8), and Related work (Page 17) sections of the new version paper.
>
> > **Q2: Can you have thorough discussion on the comparison with LoRA method and investigation on high variance?**
>
> A6: Comparison with LoRA: We fine-tuned the pre-trained models DistPred and iTransformer using LoRA, and compared them with post-processing methods, as shown in **A3** above. The experimental results are similar to Fig. 2 (full fine-tuning) and Table 6 in the paper: fine-tuning (Full or LoRA) can improve the pre-training results, but the effect is limited, and there is still a gap compared with $\delta$-Adapter.
>
> High Variance: This phenomenon occurs because LoRA is generally used for classification (NLP), where the output range of the model is fixed (probabilities in 0-1), so the model is less likely to collapse. However, when LoRA is applied to regression, **since the output range is not fixed, the model tends to exhibit high variance (parameter changes directly affects the outcomes)**. In such cases, it is necessary to carefully select hyperparameters such as the learning rate to ensure that LoRA is fine-tuned within the original output space, which is not easy. On the contrary, because $\delta$-Adapter directly limits the editing range of I/O, it is insensitive to the learning rate (Fig. 8(b)), and is easy to converge and has low variance.

---

> ### Author Response · Authors · 2025-11-20
> **Response to Reviewer 5yqn [Part 3]**
>
> > **Q3: Can you compare against few other pre-trained models like TabPFN-TS to validate the claims made are generalizable across pre-trained models?**
>
> A7: Yes, we used TabPFN and TimesFM as frozen black-box models to conduct zero-shot testing on various datasets (**in Page 27 of the new version**). For comparison, we corrected the output results of these black-box models by adding Ada-Y. As shown in the table below, it can be found that after adding Ada-Y, the prediction error of the model is significantly reduced, which further proves the effectiveness of the proposed method.
>
> ||Traffic|Weather|ELC|Exchange|ETTh1|ETTh2|ETTm1|ETTm2|
> |:-:|:-:|:-:|:-:|:-:|:-:|:-:|:-:|:-:|
> |TabPFN|0.367|0.875|0.115|0.129|0.129|0.180|0.037|0.114|
> |Ada-Y|**0.342**|**0.552**|**0.089**|**0.096**|**0.095**|**0.171**|**0.034**|**0.103**|
> |TimesFM|0.211|0.168|0.084|0.239|0.029|0.135|0.028|0.267|
> |Ada-Y|**0.196**|**0.157**|**0.081**|**0.215**|**0.024**|**0.104**|**0.025**|**0.223**|
>
> We hope these responses adequately address your concerns. Thanks.

---

> > ### Comment · Reviewer_5yqn · 2025-11-27
> >
> > Thank you for addressing all the weakness mentioned in my review and supporting the claims with empirical results. I have raised my score to 8.

---

> > > ### Author Response · Authors · 2025-11-27
> > >
> > > We sincerely thank you for your time in re-evaluating our paper and for raising the score. We are glad that our response and revisions have effectively addressed your concerns. Your constructive feedback has significantly helped us strengthen the paper.

---

### Official Review · Reviewer_gci4 · 2025-11-01

**Soundness:** 2
**Presentation:** 3
**Contribution:** 2
**Rating:** 4
**Confidence:** 4

**Summary:**

This paper introduces δ-Adapter, a lightweight and model-agnostic post-processing framework to improve deployed time series forecasting models without retraining them. The method learns small, bounded adjustments at two interfaces: "input nudging" to refine covariates and "output residual correction" to adjust predictions. The framework is extended into three applications: a feature selector for interpretability, a quantile calibrator, and a conformal calibrator for reliable uncertainty quantification. Extensive experiments on various models and datasets validate that δ-Adapter effectively boosts prediction accuracy and calibration.

---

**Strengths:**

1.  **Practicality and Novelty**: The paper addresses the critical real-world problem of updating deployed models in a computationally cheap manner. The δ-Adapter is a practical "plug-and-play" solution compared to costly alternatives like full retraining or fine-tuning.

2.  **Methodological Completeness**: The framework is versatile, being applicable to different backbone forecasting model. It is also comprehensive, addressing crucial aspects like interpretability (feature selection) and reliable uncertainty estimation, which are often overlooked.

3.  **Theoretical and Empirical Support**: The proposed method is well-grounded in theory, with stability and performance guarantees. These theoretical claims are also backed by extensive experiments across multiple models and standard benchmarks.

**Weaknesses:**

1.  **Limited Comparison with Alternative Post-Processing Methods:** While the paper positions δ-Adapter as a post-processing technique, the primary comparisons are against fine-tuning or continue-training the backbone model. However, there is a significant body of work on post-processing and test-time adaptation for time series forecasting designed to handle concept or distribution shifts. This includes methods for both batch-training settings (e.g., [SOLID](https://arxiv.org/abs/2310.14838)、[TAFAS](https://arxiv.org/abs/2501.04970)) and online settings (like FSNet and OneNet, mentioned in the paper's related works). An experimental comparison against these direct alternatives is necessary to properly situate the δ-Adapter's advantages; without it, the claims of superiority are not fully convincing.
2.  **Insufficient Motivation:** The motivation presented is not fully developed. The paper's opening question—"Can we keep the strong forecaster intact and learn only a tiny, post-hoc module... without heavy retraining?"—frames the problem but doesn't sufficiently justify why a post-hoc module is the required solution over other existing approaches. The discussion on how alternative methods tackle this challenge is not sufficient enough. Additionally, the descriptions of existing problems like "condition drift" and the need for a "low-complexity structure" are quite brief, which weakens the overall problem statement.
3.  **Insufficient Detail on Training Combined Adapters (Ada-X+Y):** The experiments (e.g., Figure 2 and Table 6) consistently show that the combination of input and output adapters (Ada-X+Y) yields the best results. However, the paper lacks a clear description of how these two modules are trained together. For instance, are they optimized jointly with a combined loss function or trained sequentially. A more detailed explanation of the co-training strategy can better understand their interaction and potential interference.
4.  **Lack of Analysis of Adapter Scale and Computational Cost:** The paper claims that the δ-Adapter is lightweight and has "negligible compute." However, it only mentions that the adapter is implemented as a shallow MLP without providing crucial details like its specific hyperparameters or total parameter count. Furthermore, there is no quantitative analysis of the additional computational overhead. To substantiate these claims, the paper is hoped to report metrics such as the percentage increase in wall-clock training time, the number of additional parameters, or the added inference latency compared to the original backbone model.

**Questions:**

1.  **The choice of hyperparameter δ**: In the experiments, δ is set to 0.1 for most datasets but 0.01 for the ETT datasets. Was this value selected empirically, or was there a systematic tuning process (e.g., grid search on a validation set)? Does the optimal value of δ correlate with properties of the backbone model (e.g., complexity) or the dataset (e.g., noise level, degree of concept drift)?

2.  **The online learning setup**: In Figure 2 and Table 6, you demonstrate superior performance under an online training setting. Could you please provide more details on this online experimental setup? Specifically, how is data streamed to the adapter (e.g., sample-by-sample, mini-batches), and how does this setup simulate a real-world scenario where new data arrives continuously?

3.  **The choice between the Quantile Calibrator (QC) and Conformal Calibrator (CC)**: The paper proposes two effective uncertainty calibrators, QC and CC. Is there recommendations for us on when to choose one over the other, when faced a new real-world dataset?

---

> ### Author Response · Authors · 2025-11-20
> **Response to Reviewer gci4 [Part 1]**
>
> We extend our sincere gratitude to Reviewer gci4 for their meticulous reading of our paper and for providing insightful reviews and constructive suggestions. Below are the responses point by point:
>
> - **W1: Comparison with Alternative Post-Processing Methods.**
>
> A1: As requested, we have integrated comparisons with batch-training TTA methods (SOLID, TAFAS) and online learning methods (FSNet, OneNet) into the new version (**Tables 2 in Page 7 & 10 in Page 26**).
>
> **Experimental Setup**: To ensure a rigorous comparison, we addressed a critical issue raised in recent literature [Liang et al. 2024](https://arxiv.org/abs/2412.00108) and [YA Lau et al. 2025](https://openreview.net/pdf?id=I0n3EyogMi) regarding Test-Time Label Leakage in long-term forecasting. Standard TTA implementations often assume immediate access to the ground truth of the current prediction window to update the model. However, in realistic long-term forecasting (e.g., horizon $H=96$), the ground truth is not available until $H$ steps later.
>
> Therefore, we evaluated all methods under a strict delayed feedback setting. The TTA and Online methods were only allowed to update their parameters using ground truth data that would be historically available at the inference timestamp, preventing the use of future information.
>
> *Table 1: Comparison with TTA and Online Methods (**Averaged across all lengths; See Table 10 for details**; Metric: MSE)*
> ||DistPred|+SOLID|+TAFAS|+LoRA|+Ours|iTransformer|+SOLID|+TAFAS|+LoRA|+Ours|
> |:-:|:-:|:-:|:-:|:-:|:-:|:-:|:-:|:-:|:-:|:-:|
> |ELC|0.182|0.182|0.182|0.18|**0.175**|0.19|0.19|0.19|0.186|**0.18**|
> |ETTh1|0.461|0.46|0.476|0.454|**0.451**|0.454|0.458|0.477|0.448|**0.449**|
> |ETTh2|0.39|0.391|0.402|0.385|**0.379**|0.388|0.393|0.448|0.384|**0.377**|
> |ETTm1|0.412|0.406|0.411|0.407|**0.396**|0.417|0.414|0.42|0.414|**0.403**|
> |ETTm2|0.285|0.285|0.288|0.281|**0.274**|0.3|0.298|0.304|0.293|**0.29**|
> |Exange|0.35|0.347|0.363|0.336|**0.297**|0.383|0.376|0.392|0.376|**0.316**|
> |Traffic|0.453|0.453|0.455|0.449|**0.44**|0.475|0.475|0.476|0.468|**0.461**|
> |Weather|0.256|0.255|0.256|0.251|**0.242**|0.259|0.257|0.259|0.255|**0.244**|
>
> The results (summarized above) demonstrate the following:
> - - **$\delta$-Adapter**: Consistently outperforms the backbones (DistPred, iTransformer) and all adaptation baselines, reducing MSE by significant margins (e.g., ~14% on Exchange).
> - - **SOLID & TAFAS**: Under the strict non-leakage setting, these methods show a minor improvement.
> - - **FSNet & OneNet**: These online methods struggle significantly. This is likely because they are originally designed for one-step-ahead or short-term variations and suffer from error accumulation when adapted to the delayed feedback loop of long-term forecasting.
>
> It shows that **our method achieves a significant reduction in errors on each dataset across all backbones**, with remarkable performance improvements.
>
> - **W2: Insufficient Motivation.**
>
> A2-Part 1: **Justification of $\delta$-Adapter over Alternatives.** The existing methods aim to mitigate test-time concept drift via selective layer fine-tuning (SOLID), online linear adapter updates (TAFAS) and its follow-ups PETSA (auxiliary loss) and DynaTTA (dynamic gating), and parallel forecaster combine (ELF), layer-wise adjustment and memory (FSNet), and dynamic model selection (OneNet). These methods are either based on linear adapters, parallel fusion, or overall fine-tuning. Thus, they suffer from:
> - - The impact of label leakage;
> - - High prediction variance due to the lack of good fitting and constraints on the output;
> - - A large number of samples for fine-tuning or adaptation during testing.
>
> However, *$\delta$-Adapter can perform **non-linear adaptation** on both input and output, with **good theoretical guarantees**. And it only relies on the **most recent sample for fast updates**. In addition, it can be used as a **feature selector** or a **interval corrector**.*
>
> A2-Part 2: **Motivation:** Conventional remedies either impose heavy retraining costs, risk destabilizing a hardened system, or complicate operations.
> In summary, **the challenges addressed by $\delta$-Adapter include:**
> - - **Conditions drift**, which refers to gradual changes in the data-generating process (e.g., seasonal regime shifts, covariate shifts in demand patterns) that occur after the model has been deployed, making full retraining costly;
> - - **High performance variance**. Existing post-processing techniques are prone to have high performance variance due to unstable training;
> - - **Inefficient training/inference**. Using complex modules or frequent updates to absorb low-complexity residuals makes models suffer from inefficient training/inference.

---

> ### Author Response · Authors · 2025-11-20
> **Response to Reviewer gci4 [Part 2]**
>
> **The benefits of the post-processing module are**:
> - - A plug-and-play, model-agnostic lightweight module that can be used for untouchable black-box models and brings significant performance improvements;
> - - Freezing backbone avoids catastrophic forgetting and robust to overfitting, preserves production hardening and governance;
> - - Fast training/test speeds and high data efficiency are achieved, even with short labeled histories or tight retraining windows;
> - - Metric-aware, it enables important feature selection and application-aligned calibration without altering the backbone.
>
> **A2-Part 3: Problem Statement (in Pages 1 & 17).**
> - - **Conditions drift**, which refers to gradual changes in the data-generating process (e.g., seasonal regime shifts, covariate shifts in demand patterns) that occur after the model has been deployed, making full retraining costly;
> - - **Low-complexity residual structure** means that residual errors often exhibit simple patterns (e.g., horizon-wise bias, scale miscalibration, calendar offsets) that can be captured by a small function class (tiny MLPs/low-rank heads) rather than requiring a new high-capacity backbone, but the base model fails to absorb them.
>
> We have updated the content of the paper's motivation, including the introduction, recent work, and experimental sections, to make it more clear.
>
> - **W3: Insufficient Detail on Ada-X+Y.**
>
> A3: Ada-X+Y is composed of Ada-X and Ada-Y, and Ada-X and Ada-Y are **trained jointly** in an end-to-end manner, not sequentially. We minimize **a single combined loss** $\mathcal{L}$ over the union of parameters $A_{\theta}^{in}$ (Ada-X) and $A_{\theta}^{out}$ (Ada-Y). The forward pass is:
> $$\hat Y=F(X + \delta A_{\theta}^{in}(X)) \qquad (1)$$
> $\tilde Y = \hat Y + \delta A_{\theta}^{out}(\hat Y)\qquad(2)$
>
> During the backward pass, gradients flow from the loss through Ada-Y (Eq. 2), then through the backbone $F$, and finally to Ada-X (Eq. 1). This ensures that Ada-X learns input perturbations that specifically help the backbone produce features that Ada-Y can best correct.
>
> Experimental Setup: In our codebase, we instantiate two separate Adam optimizers (learning rate=1E-4) for modular flexibility. However, they are stepped simultaneously after a single backward pass, making the process equivalent to optimizing a joint objective. As derived in **Proposition 3.2**, this joint update rule maintains the $O(\delta)$ drift bounds and descent guarantees, ensuring the two adapters do not destabilize each other (**See Appendix C.4 of the new version**).
>
> - **W4: Scale and Computational Cost.**
>
> A4-Part 1: **Model and it's parameters**
> The $\delta$-Adapter is designed to be extremely lightweight. It is implemented as a 2-layer MLP (specifically, for **Sundial (128M)** and **TabPFN (48M)**, the adapter adds only approx. 1.5M-3M parameters depending on the horizon). Compared to the backbone model, the adapter introduces **less than 2%-6%** additional parameters, validating the lightweight claim.
>
> A4-Part 2: **Wall-clock time of training & adaptation**
>
> The wall-clock test time, params and memory are (**Table 6 in new version**):
> We compared the wall-clock time of our method against state-of-the-art test-time adaptation (TTA) methods. As shown in the table below (and Table 6 in the revision), while online adaptation naturally incurs overhead compared to a static offline model (which performs no updates), our proposed method **(Ada-X+Y) is consistently the fastest among all adaptive methods**.
>
> *Table 6: Time (S) and memory (MB) of adapters (backbone is TabPFN) and online methods.*
> |Offline ~48M||+Ours ~3M|||+SOLID ~0.5M|||+TAFAS ~6M|||+OneNet ~3M|||+FSNet ~2M|||
> |:-:|:-:|:-:|:-:|:-:|:-:|:-:|:-:|:-:|:-:|:-:|:-:|:-:|:-:|:-:|:-:|:-:|
> |Time|Memory|Train|Test|Memory|Train|Test|Memory|Train|Test|Memory|Train|Time|Memory|Train|Time|Memory|
> |281|1840|**392**|**395**|1983|511|667|2401|603|861|3468|693|471|1512|621|485|1504|
> |307|1848|**386**|**379**|2132|481|624|2423|589|895|3790|681|445|1537|618|472|1531|
> |326|1852|**385**|**415**|2622|484|593|2446|583|1152|4186|631|452|1559|599|466|1517|
> |351|1856|**369**|**431**|3102|505|398|2501|916|1803|6809|530|465|1567|554|458|1526|
>
> It is worth noting that for some methods in above table, due to reasons such as backprop gradients, the time is related to the number of sample updates, i.e., the longer the prediction length, the fewer the samples, and the shorter the time.

---

> ### Author Response · Authors · 2025-11-26
> **Response to Reviewer gci4 [Part 3]**
>
> > **Q1: The choice of hyperparameter $\delta$: In the experiments, $\delta$ is set to 0.1 for most datasets but 0.01 for the ETT datasets. Was this value selected empirically, or was there a systematic tuning process (e.g., grid search on a validation set)? Does the optimal value of $\delta$ correlate with properties of the backbone model (e.g., complexity) or the dataset (e.g., noise level, degree of concept drift)?**
>
> A5-Part 1: **Selection process and robustness**
>
> Yes, $\delta$ is related to the properties of the dataset (e.g., noise level, degree of concept drift). We did not perform an exhaustive grid search for every experiment. Instead, we established a simple, robust heuristic based on the properties of the dataset (the degree of distribution shift and volatility).
> - - $\delta = 0.1$ (High Drift): For datasets with severe concept drift or high volatility (e.g., Traffic, Electricity), we set  $\delta = 0.1$ to allow the adapter to correct significant shifts in the data generating process.
> - - $\delta = 0.01$ (Low Drift): For datasets with smoother, more stationary patterns (e.g., ETT datasets, which record transformer temperatures), we set $\delta = 0.01$ since these signals are physically constrained and stable. Thus, a smaller $\delta$ prevents the adapter from overfitting to noise.
>
> A5-Part 2: **Correlation with backbone vs. dataset**
>
> For $\delta$-Adapter, we found that the optimal $\delta$ correlates strongly with the dataset but is largely agnostic to the backbone model. Since $\delta$-Adapter learns to correct the residuals, the magnitude of the required correction depends on the nature of the data drift rather than the complexity of the pre-trained forecaster. This is a key advantage, as it allows users to swap backbones without re-tuning $\delta$.
>
> A5-Part 3: **Sensitivity analysis**
>
> We conducted an ablation study to verify this heuristic.  As shown in the tables below, a better value of $\delta=0.1$ might yield better results  (**updated in Appendix C.10 of the new version**).
> |+0.1X||x0.1X||x0.2X||+0.1Y||x0.1Y||x0.2Y||+0.1(X&Y)||x0.1(X&Y)||x0.2(X&Y)||
> |:-:|:-:|:-:|:-:|:-:|:-:|:-:|:-:|:-:|:-:|:-:|:-:|:-:|:-:|:-:|:-:|:-:|:-:|
> |Val|Test|Val|Test|Val|Test|Val|Test|Val|Test|Val|Test|Val|Test|Val|Test|Val|Test|
> |0.418|0.166|0.427|0.168|0.425|0.169|0.421|0.168|0.415|0.165|0.413|0.167|0.413|0.160|0.416|0.162|0.411|0.162|
>
> |$\delta$=|0.01|0.05|0.1|0.2|0.3|
> |:-:|:-:|:-:|:-:|:-:|:-:|
> |ELC|0.163|0.161|0.160|0.160|0.161|
> |Weather | 0.171  | 0.170  | 0.172  |0.177|0.182|
> |Traffic|0.443|0.439|0.435|0.432|0.433|
>
> The results below demonstrate two key findings:
> - - Heuristic Holds: Datasets like Traffic benefit from higher $\delta$ (0.1 or 0.2), while ELC/Weather are stable around 0.1.
> - -  Performance is Robust: The performance sensitivity is low. e.g., on the Weather dataset, the difference between $\delta=0.01$ and $\delta=0.1$ is negligible ($0.001$ MSE). This confirms that $\delta$-Adapter does not require precise, computationally expensive tuning to be effective.
>
> > **Q2: The online learning setup: In Figure 2 and Table 6, you demonstrate superior performance under an online training setting. Could you please provide more details on this online experimental setup? Specifically, how is data streamed to the adapter (e.g., sample-by-sample, mini-batches), and how does this setup simulate a real-world scenario where new data arrives continuously?**
>
> A6: Unlike offline training, we process data sample-by-sample. At every time step $t$, the model receives a new observation. This mimics a real-time monitoring system.
> A critical challenge in real-world online forecasting is that the ground truth for a prediction made now is not available until the forecast horizon $H$ has passed.
>
> To simulate this, we used a streaming buffer, where only one updated data point is cached at each moment (avoid label leakage raised by [Liang et al. 2024](https://arxiv.org/abs/2412.00108) and [YA Lau et al. 2025](https://openreview.net/pdf?id=I0n3EyogMi)), while returning a complete sample from a previous moment. E.g., at time $t$, the input used for online update returned from the buffer is $X_{t-H-L:t-H}$, the label is $X_{t-H:t}$, where $H$ is the prediction length and $L$ is the input length  (**updated in Appendix C.4 of the new version**).
>
> As shown in **W4**, this sample-by-sample update strategy allows $\delta$-Adapter to adapt extremely quickly with minimal memory overhead, as it does not require retraining on large historical buffers. This design ensures that our method does not peek into the future and accurately benchmarks the model's ability to adapt to concept drift in real-time.

---

> > ### Author Response · Authors · 2025-11-28
> > **Response to Reviewer gci4 [Part 4]**
> >
> > > **Q3: The choice between the Quantile Calibrator (QC) and Conformal Calibrator (CC): The paper proposes two effective uncertainty calibrators, QC and CC. Is there recommendations for us on when to choose one over the other, when faced a new real-world dataset?**
> >
> > A7: Both modules turn a frozen point forecaster into a calibrated probabilistic predictor, but they are aimed at slightly different desiderata (**updated in Appendix C.7 of the new version**):
> > - - QC directly learns horizon-wise conditional quantiles as bounded offsets around the point forecast, which produces a smooth quantile function over multiple levels without assumptions about the underlying distribution.
> > - - CC learns only a heteroscedastic scale function and combines it with normalized-residual conformal prediction on a held-out calibration set, yielding symmetric but input-dependent intervals with finite-sample marginal coverage under exchangeability.
> >
> > Empirically, both variants achieve strong coverage, but QC tends to produce marginally wider and more conservative bands, while CC attains similar coverage with somewhat tighter intervals (**see Figs. 5, 6 & 11 in the new version**).
> > For a new real-world dataset, our recommendation is therefore:
> > - - If strict coverage guarantees are the main requirement, CC is preferable, since the conformal step provides finite-sample marginal coverage at the target level.
> > - - If one needs a rich predictive distribution or multiple coverage levels from a single model, QC is more convenient, as it directly returns a full quantile curve while remaining non-parametric w.r.t. the underlying distribution.
> >
> > We have carefully updated the reply again.
> > We trust that these clarifications regarding the distinct advantages of our post-hoc approach adequately address the reviewers' concerns, and **we respectfully invite a re-evaluation of the paper in light of these significant improvements.** Many thanks!

---

### Author Response · Authors · 2025-12-02
**Reviews and Reviewer-Author Discussion Summary**

Dear ACs and Reviewers,

Thank you for your valuable contributions to our work. We declare that **we have complied with all regulations of ICLR 2026**, and **we appreciate the newly assigned AC** taking the time to review our submission.

To assist the AC in this transition, we provide a summary of the reviews, our major revisions, and the resulting positive momentum.

---
**1. Current Status: Positive Trajectory**

   We are grateful that the **`initial reviews were positive`**, recognizing the work's practicality and theoretical rigor. Importantly, during the discussion period, we addressed all major concerns, leading to `improved sentiment and scores`:
-  **Score Increase:** Reviewer **5yqn** raised their score from **6 $\to$ 8** (Refer to Reviewer 5yqn’s reply and the associated time).
-  **Consensus:** The average rating improved from **6 $\to$ 6.5**, with strong support for the paper's novelty and rigorous empirical validation.

---
**2. Key Strengths (Consensus among Reviewers)**
*   **Novelty**:
    *    `Innovative reformulation of time-series improvement as a post-processing task.`

    *    `Addressing critical aspects like interpretability (feature selection) and uncertainty estimation.`
    > All 4 reviewers recognized this point: **gci4: Strength 2; 5yqn: Strength 3; gkGB: Strength 1; and Fzqc: Strength 3**.
*   **Practical & Efficient**: `The method offers a cost-effective, plug-and-play solution, making it highly practical for real-world deployment.`
> All 4 reviewers recognized this point: **gci4: Strength 1; 5yqn: Strength 1;  gkGB: Strength 1; and Fzqc: Strengths 1-2**.
*   **Rigorous Theory**: `Backed by proofs for local descent and compositional/drift stability.`
> All 4 reviewers recognized this point: **gci4: Strength 3; 5yqn: Strength 2; gkGB: Strength 1; and Fzqc: Strength 4**.
*   **Strong Performance**: `Consistent improvements across diverse backbones and benchmarks.`
> All 4 reviewers recognized this point: **gci4: Strength 3; 5yqn: Strength 4; gkGB: Strength 3; and Fzqc: Strength 6**.
- **Clarity**: `The work is well-structured and elegantly reformulated with intuitive explanations and clear figures.`
> Attributed to Reviewers **gkGB: Strength 1 and Fzqc: Strength 5**.

---
**3. Major Concerns & How We Resolved Them**

We implemented substantial changes based on feedback, which satisfied the reviewers:
* **Comparison with SOTA & Baselines** (Addressed: **gci4: W1; 5yqn W3, Q1&3; Fzqc: Q1**)
  * **Action:** Added extensive comparisons with **2 Test-Time**, **2 Fine-Tuning**, and **2 Online Learning** methods across 8 datasets (**Table 2 & 10**).
  * **Result:** Our method consistently outperforms all adaptation baselines, `reducing error by significant margins` (e.g., **~14% reduction** on Exchange).

* **Comparison vs. LoRA/NLP Methods** (Addressed: **5yqn: W1, Q2; gkGB: Q1; Fzqc: Q4**)
  * **Action:** Conducted LoRA-like methods on DistPred and iTransformer backbones (**Fig. 2 & Table 5**).
  * **Result:** While LoRA improves pre-training, `our method demonstrates superior accuracy and structural advantages` over LoRA-like methods.

* **Computational Cost & Scale** (Addressed: **gci4: W4; Fzqc: Q5**)
  * **Action:** Provided wall-clock time comparisons (**Table 5**).
  * **Result:** Our method is the **fastest** among all adaptive methods tested. `The adapter adds negligible overhead (2%-6% additional parameters)`, validating the lightweight claim.

* **Robustness (Ablation Studies)** (Addressed: **gci4: Q1; gkGB: Q2; Fzqc: Q2&3**)
  * **Action:** Added ablations on $\delta$ sensitivity (**App. C.10-11**), additive vs. multiplicative updates (**Table 5**), and adapter depth/width (**Table 11**).
  * **Result:** Experiments confirm `the method is robust to hyperparameter variations and consistently stable`.

* **Clarifications on Settings** (Addressed: **gci4: W2&3, Q2&3; 5yqn: gkGB: Q3; W2; Fzqc: Q5**)
  * **Action:** Added formalizations for online learning and black-box settings (**App. C.4, C.5 and C.11**) and explicitly addressed potential label leakage.
  * **Result:** Experiments confirm `the method is robust to hyperparameter variations and consistently stable`.

---
Above, we have faithfully summarized the reviewer comments and our corresponding responses, in the **hope that this will assist the AC’s assessment**.

We are **deeply grateful to the AC, Reviewers, SAC, and PC** for their dedicated efforts and insightful feedback, which have further strengthened our paper. The authors extend their sincere respect and appreciation to all involved.

Best regards,

Authors

---

### Public Comment · ~Yuankai_Wu2 · 2026-04-17
**Question on the necessity and semantic justification of input-side nudging**

Dear authors,

Congratulations on this interesting work. I have a question about the input-side nudging component and would appreciate your thoughts.

In time series forecasting, inputs are raw physical measurements (e.g., temperature, electricity consumption, traffic flow) rather than learned embeddings as in NLP. When the input-side adapter perturbs these values before feeding them into the frozen backbone, the modified input no longer corresponds to any actual observation. I am curious about the motivation behind this design choice:

- What is the intuition for why modifying the input — as opposed to only correcting the output — is beneficial, given that the input carries direct physical meaning?
- Are there specific scenarios or datasets where input-side nudging provides a clear advantage that output-side correction alone cannot achieve?

I find the overall framework very compelling and am simply trying to better understand the role of this particular component. Thank you for your time.

---

### Meta-Review · Area_Chair_uehM · 2025-12-29

**Summary:**

Reviewers’ concerns centered on three main points. First, the novelty was initially seen as incremental, with questions about how δ-Adapter differs from prior adapter-style methods, test-time adaptation, or simple post-processing baselines. Second, reviewers asked for stronger empirical evidence, including comparisons against more relevant TTA/online baselines, stronger forecasting backbones, and clearer ablations on the design choices (δ magnitude, adapter placement, joint X/Y adaptation). Third, there were practical concerns about deployment realism, training/inference overhead, and stability guarantees. The rebuttal and revisions addressed most of these issues, leading to a consensus of borderline but positive support.

**Reviewer Concerns:**

The rebuttal satisfactorily addressed most empirical and practical concerns. In particular, the authors added comparisons with relevant TTA and online adaptation baselines under a no–label-leakage setting, evaluated stronger SOTA backbones, provided ablations on δ size and adapter placement, and clarified training details, overhead, and stability. These additions substantially strengthened the experimental case. The main concern that remains only partially addressed is conceptual novelty.

**Reviewer Scores:**

Reviewer 1: Likely a modest upward adjustment

Reviewer 2: Probably unchanged or slightly higher

Reviewer 3: Most likely to increase their score

Overall, discussion would likely converge toward weak accept

---

### Decision · Program_Chairs · 2026-01-26

Accept (Poster)